# A hyperspectral and multi-angular synthetic dataset for algorithm development in waters of varying trophic levels and optical complexity

Jaime Pitarch[1], Vittorio Ernesto Brando[1,2]

[1]Consiglio Nazionale delle Ricerche (CNR), Istituto di Scienze Marine (ISMAR), Via Fosso del Cavaliere 100, 00133 Rome, Italy
[2]Commonwealth Scientific and Industrial Research Organisation (CSIRO), Environment, Canberra, Australia

*Correspondence to*: Jaime Pitarch (jaime.pitarch@cnr.it)

**Abstract.** This data paper outlines the development and structure of a synthetic dataset (SD) within the optical domain, encompassing inherent and apparent optical properties (IOPs-AOPs) alongside associated optically active constituents (OACs). The bio-optical modeling benefited from knowledge and data accumulated over the past three decades, resulting on a comprehensive dataset of in situ IOPs, including diverse water typologies, and enabling the imposition of rigorous quality standards. Consequently, the bio-optical relationships delineated herein represent valuable contributions to the field.

Employing the Hydrolight scalar radiative transfer equation solver, we generated above-surface and submarine light fields across the specified spectral range at a “true” hyperspectral resolution (1 nm), covering the ultraviolet down to 350 nm, therefore facilitating algorithm development and assessment for present and forthcoming hyperspectral satellite missions. A condensed version of the dataset tailored to twelve Sentinel-3 OLCI bands (400 nm to 753 nm) was crafted. Derived AOPs encompass an array of above- and below-surface reflectances, diffuse attenuation coefficients, and average cosines.

The dataset is distributed in 5000 files, each file encapsulating a specific IOP scenario, ensuring sufficient data volume for each water type represented. A unique feature of our dataset lies in the calculation of AOPs across the complete range of solar and viewing zenith and azimuthal angles as per the Hydrolight default quadrants, amounting to 1300 angular combinations. This comprehensive directional coverage caters to studies investigating signal directionality, previously lacking sufficient reference data. The

dataset is publicly available for anonymous retrieval via the FAIR repository Zenodo at https://doi.org/10.5281/zenodo.11637178 (Pitarch and Brando, 2024).

## 1. Introduction and review

### 1.1 Background

Marine optics studies the light measured by an optical radiometer, whether installed in the water or above the surface. The optical signal is conveniently formulated in terms of apparent optical properties (AOPs), which are normalized quantities, less dependent on the intensity of the incident light than the radiances or irradiances from which they originate. The most notable AOP is the remote-sensing reflectance ($R_{rs}$), defined as the water-leaving radiance $L_w$ per unit of above-water planar downwelling irradiance $E_s$. Other quantities like diffuse attenuation coefficients and average cosines find applications in marine optics (Mobley, 1994).

AOPs are used to retrieve the concentrations of optically active water constituents (OACs). Such OACs have historically been marine phytoplankton and other suspended and dissolved substances. Phytoplankton is typically quantified in terms of the chlorophyll concentration (C). All the other materials suspended in the water can be grouped in the non-algal particles (NAP), quantified by their concentration (N). Dissolved substances are not commonly given in terms of mass concentration units, but in terms of the absorption coefficient spectrum, commonly at 440 nm (Y, or $a_g(440)$).

It is possible to develop empirical algorithms to invert any of the OACs from measured AOPs by developing statistical relationships from matched AOP and OAC data (IOCCG, 2006). This approach, although sometimes operationally robust and mechanistically meaningful, hampers progress in understanding the optical influence of OACs, which is given by the inherent optical properties (IOPs), namely the absorption and scattering coefficients. As such, the IOPs can be mathematically linked to the OACs and therefore be a mathematical bridge between the AOPs and the OACs (Mobley, 1994).

The OACs are the independent variables that drive the generation of a synthetic dataset. They can be a single quantity like C (IOCCG, 2006;Loisel et al., 2023) or, alternatively, a triplet formed by C, N and Y

(Nechad et al., 2015). The first case is typically chosen for open sea conditions, whereas the second is the choice for optically complex waters.

In either case, to model the radiant field, relationships between the IOPs and the OACs must be set. Relationships between C and IOPs have been already studied for decades (Bricaud et al., 1998;Loisel and Morel, 1998;Morel and Maritorena, 2001). Much less is known about N and Y, and in particular, in optically complex waters, where there is no known relationship between the OACs, but also their bio-optical properties are much more variable. Nevertheless, there are notable bio-optical studies in Australian waters (Blondeau-Patissier et al., 2009;Cherukuru et al., 2016;Blondeau-Patissier et al., 2017), European waters (Tilstone et al., 2012;Martinez-Vicente et al., 2010;Astoreca et al., 2012), South-African lakes (Matthews and Bernard, 2013) and North-American coastal waters (Aurin et al., 2010;Le et al., 2013;Le et al., 2015).

In order to develop new bio-optical relationships, there is a need for large datasets of concomitant OAC-IOP data across a range of data values, seasons and geographical locations, with fully characterized uncertainties, but despite a broader accessibility to field- and laboratory-based IOP instrumentation, data availability and quality is not what was expected twenty-five years ago, when instrumentation became commercially available. IOP measurements are scarce, strongly concentrated in some areas and without characterized uncertainties.

Studying the relationships between the IOPs and AOPs allows to build semianalytical models of ocean color: these are simplified algebraic expressions of a desired AOP as a function of the IOPs, and they are needed to make retrieval of IOPs from AOPs feasible. Given the absence of publicly available matched IOP-AOP data across a range of water types, and with characterized uncertainties, it has been a common choice to develop synthetic datasets (SDs) for optical studies (IOCCG, 2006;Nechad et al., 2015;Loisel et al., 2023). SDs fill the gaps in the data ranges, and their IOP-AOP relationships can be considered error-free, as they are derived from the solution of the radiative transfer equation, which has solid physical foundation. As such, SDs are very powerful to develop simplified IOP-AOP relationships. In a pioneering work, Gordon et al. (1988) proposed that the underwater irradiance reflectance (R) could be modelled as a second-degree polynomial of a parameter "X" that, translating to today's notation, was equivalent to $\frac{b_b}{a+b_b}$. They used Monte Carlo modelling to generate a synthetic dataset of matched IOPs and the irradiance

reflectance R. This approach has been followed since then by many authors, proposing other analytical expressions and changing the fitted variables, but essentially the approach remains the one by Gordon, with variations. If different sun-view geometries are considered, the bidirectional aspects of the AOPs can be studied (Morel and Gentili, 1993, 1996;Morel et al., 2002;Park and Ruddick, 2005;Lee et al., 2011).

Other applications of SDs are related to algorithm development and testing. Matched values of the variable of interest and the input data to be retrieved from (usually an AOP) are used as training data to develop an algorithm. This can be from a simple analytical expression, like the retrieval of non-water absorption at a green band from $R_{rs}$ in the quasi-analytical algorithm (Lee et al., 2002). At the other end of the algorithm complexity, Doerffer and Schiller (2007) elaborated their MERIS Case 2 water algorithm using an ad-hoc synthetic dataset.

Hyperspectral datasets can be used to develop inherently hyperspectral algorithms, and they are also useful to study how much information is embedded in some key bands of multispectral sensors. In this respect, Talone et al. (2024) used a preliminary version of this SD to propose a hyperspectral $R_{rs}$ reconstruction scheme from AERONET-OC data, in order to validate satellite derived hyperspectral radiometric products, confirming the validity in large portions of the visible spectrum with constrained uncertainties.

### 1.2 Existing synthetic datasets

The usage of numerical models for computing light fields has been common practice for several decades already (Mobley et al., 1993). Some researchers have developed internal codes (D'Alimonte et al., 2010) while some others have released them to the public (Chami et al., 2015;Rozanov et al., 2014). By far, the most popular code in the marine optics community has been Hydrolight (formerly from Sequoia Scientific, Inc., now from Numerical Optics, Ltd.), which is available upon purchase. Its popularity is due to, on one hand, the convenient management of data input, which allows the simulation of every possible case study in ocean optics with relative ease, and the data output, which includes the full array of radiometric quantities and AOPs needed. Its prevalence in the field is such that all datasets reviewed in

this paper, as well as the one presented here, were generated with Hydrolight. It is therefore of importance that support and further development of Hydrolight is ensured for the future.

Most of previously developed were developed to fit a given investigation and were not released to the public. This article only considers those that were publicly released. Only their main characteristics will be mentioned, especially those relevant for the new synthetic dataset that we are presenting.

#### 1.2.1 The IOCCG dataset

The first and the most cited of the datasets in this small review is the IOCCG dataset (IOCCG, 2006). The release of this dataset came at a time where the study of bio-optical relationships and the development of algorithms was at its all-time peak (e.g., Twardowski et al., 2001;Loisel and Morel, 1998;Morel and Maritorena, 2001;Lee et al., 2002). It is a dataset for testing and development of in-water algorithms in open and oceanic waters. As such, the single independent variable of the IOP variability is the chlorophyll concentration (C), for concentrations between 0.03 and 30 mg m$^{-3}$. Phytoplankton absorption is the only actually measured IOP that is used, coming from a database of in situ absorption spectra ($a_{ph}$). Given a C value, a random $a_{ph}$ is chosen within the database, and it is scaled so that the scaled $a_{ph}(440)$ verifies the relationship to C given by Bricaud et al. (1995). Notably, the chosen $a_{ph}$ belongs to a subset of $a_{ph}$ spectra associated to C values within a short range of the given C. This choice implies assuming that $a_{ph}$ spectra that are related to very different concentrations are not only different in magnitude, but also in shape.

The bio-optical relationships are set after (mostly) published relationships, with the addition of some randomness, that tries to model some spread around the mean relationship, attributed to natural causes that are not captured by these average equations. While that choice is a positive feature of the dataset, many parameterizations appear arbitrary.

The volume scattering function is modelled after splitting the particulate matter in phytoplankton and all non-algal (non-pigmented) particles. The former scatters light following a Fournier-Forand phase function of fixed $B_{ph} = 0.01$, whereas the latter scatters light according to the average Petzold phase function, $B_{ph} = 0.0183$. This is identified as a major limitation for this dataset, as there are a number of concerns on the Petzold phase function that will be detailed below.

Radiances were generated from 400 nm to 800 nm every 10 nm for the nadir view direction, and for two sun zenith angles (0 and 30 º).

#### 1.2.2 The Coastcolour dataset

The Coastcolour synthetic dataset (Nechad et al., 2015) was generated in the framework of an ESA project, aimed at the evaluation of algorithms in coastal waters. The project included the compilation of large amounts of in situ data, but the patchiness in the geographical and data range distributions and the disparity of measurement techniques, without quantified uncertainties, made evident the need of a synthetic dataset that focused in such areas and associated data ranges.

The dataset is driven by three OACs: phytoplankton, NAP and CDOM. 5000 triplets of their respective concentrations (C,N,Y) were randomly generated. Although not documented in their paper, these three constituents show some degree of linear crossed correlation, which is seen in in situ datasets when these variables span across a large range. This choice also mechanistically avoids the generation of many unrealistic $R_{rs}$ spectra coming from unrealistic (C,N,Y) triplets. The OACs are related to the IOPs according to some bio-optical relationships. The non-water substances were divided into phytoplankton, “mineral particles” and CDOM. This, in principle, ignores the contribution of non-algal particles of biological origin, but in practice, their “mineral particles” compartment de facto stands for “non-algal particles”.

Bio-optical modelling relationships are based on average parameters and regression equations from literature, ignoring the natural variability. For example, phytoplankton absorption was modelled by simply applying the average “A” and “E” power law coefficients by Bricaud et al. (1995) for a given chlorophyll, which ignores phytoplankton diversity and makes all 5000 modelled $R_{rs}$ to have the same average pigment features. Furthermore, all spectral slopes as well as the “mineral particles” specific scattering coefficient are set constant. Overall, these bio-optical choices create an optical uniformity that results in artificially tight relationships between various IOPs or between IOPs and AOPs, as well as their ratios. This bio-optical modelling can potentially mislead the users about the performance of any algorithm that is evaluated.

The dataset delivers the absorption coefficient divided in the total non-water component and the phytoplankton absorption. To separate CDOM and NAP absorption, the users need to generate CDOM spectra with the reported value at a given wavelength and the spectral slope.

Following the IOCCG approach, angular scattering was modelled by assuming a Fournier-Forand phase function for phytoplankton and the average Petzold phase function for NAP, with fixed backscattering ratios for both.

AOPs were generated with Hydrolight from 350 nm to 900 nm every 5 nm, for the sun zenith angles 0, 40º and 60º, and the single nadir-viewing angle for radiances.

#### 1.2.3 Loisel's dataset

Loisel's dataset (Loisel et al., 2023) is mainly characterized by its intention to compensate the disproportionate in situ data density from coasts and continental shelfs respect to the open oceans, which instead represent a much larger area. According to them, this issue may have a biasing effect when synthetic datasets are used to develop optical algorithms based on AOP vs. IOP relationships, especially when the underlying goal is to represent a broad range of IOPs encountered within the global ocean. In this regard, satellite-retrieved IOPs over the global oceans were organized in histograms, which were used as guides to "trim" the histograms of the in situ data, so that the data distributions in the dataset would closely match the global ones.

IOP variability is driven by chlorophyll concentration only. Bio-optical modelling follows the IOCCG approach with modifications, thus choosing phytoplankton from a pool of real spectra, and giving some randomness to the relationships to mimic the bio-optical variability found in nature. The CDOM and NAP spectral slopes were given random values within a large uniform distribution, which might have been constrained with available in situ data pools. Angular scattering of phytoplankton was modelled with a fixed Fournier-Forand phase function of $B_{ph} = 0.01$. There is, however, evidence (Whitmire et al., 2010) that $B_{ph}$ varies across an order of magnitude. In Hydrolight, $B_{ph}$ is used to choose the phase function, which, for a given $b_{ph}$, implicitly determines $b_{b,ph}$ and therefore, the amplitude of the signal. This detail is important when one seeks to replicate relationships of $b_{bp}$ to other IOPs that are found in measured data.

NAP scattering was modelled as a power law. Its angular scattering incorporates one innovation respect to the previous datasets by dropping the Petzold phase function and using instead a Fournier-Forand function of $B_{NAP} = 0.018$, with such $B_{NAP}$ close to the average Petzold value, but with a more realistic angular variation.

Output AOPs are given between the range 350 nm – 750 nm in steps of 5 nm. Several versions of the dataset are available for various combinations of inelastic scattering being or not considered. Notably, this dataset provides the data output at several depths. Simulations are made for the sun zenith angles 0, 30º and 60º, and the single nadir-viewing angle for radiances. All data is compiled in a single netCDF file for each type of simulation.

### 1.3 Creating a new dataset

The brief review of existing datasets has identified limitations in bio-optical modeling, emphasizing the critical need for meticulous refinement in order to derive meaningful radiance outputs from radiative transfer simulations. Such issues can be summarized in:

(1) Overly simplified bio-optical parameters: spectral slopes, specific absorption or scattering at a reference wavelength, are often set as static values, typically derived from averaging datasets, thereby masking the optical diversity inherent within them. In this new dataset, we address this limitation by considering the variability of each optical parameter across available datasets and exploring their prediction as a function of other parameters.

(2) An absence of constraints between absorption and scattering for a given water constituent (OAC) such as phytoplankton or non-algal particles (NAP): it is evident that absorption and scattering should exhibit statistical correlations due to their association with the same type of particles, but it seems the rule that both properties are modelled independently, potentially resulting in absorption-scattering pairs that do not accurately reflect the characteristics of naturally occurring particles. In this dataset, we address this issue by leveraging in-situ data to constrain the modeling of both phytoplankton and NAP. This approach ensures that the corresponding absorption-scattering pairs align with all experimental evidence in statistical terms.

(3) Re-use of bio-optical relationships: a published relationship between two quantities is applied to different ones. For example, the average relationship between chlorophyll and particle scattering by Loisel and Morel (1998) has been used to model phytoplankton scattering, which is only a fraction of the total scattering.

(4) Limited validation of bio-optical models: using accessible in situ data is crucial. With new open-access datasets, there arises an opportunity to assess historical bio-optical relationships while also fostering the development of new ones. To our opinion, such data has not been yet fully utilized.

(5) Limited spectral coverage of the blue-UV: in view of present and future satellite missions, it is desirable to generate datasets that at least cover the range from 350 nm.

(6) Limited directional AOP output: published datasets focus on the nadir viewing direction, for a few sun zenith angles. However, the light field is inherently directional, and ignoring directionality introduces errors in remote sensing algorithms. Here, we aim at generating a fully directional dataset, accounting for all possible sun and view geometries, in view of optical studies that address the problem of directionality.

## 2. In situ data and bio-optical modelling

### 2.1 In situ data

#### 2.1.1 Phytoplankton absorption

Phytoplankton absorption $a_{ph}$ is the only IOP that is not modelled as a simple analytical function due to its complex spectral shape, which determines the spectral features of derived radiometric variables. For this reason, it is important to select high-quality $a_{ph}$ data. Since the purpose of these data is to feed the simulations, they do not need to be geo-referenced nor matched to any other variable. However, it was required that data was collected at or close to the surface, as bio-optical relationships involving phytoplankton seem to vary depending on the layer (Bricaud et al., 1995;Loisel and Morel, 1998). In terms of spectral range, a condition for data selection was that data should be given from 350 nm to 800 nm, which was a quite limiting requirement for the lower limit, as in most cases, $a_{ph}$ is provided down to 400 nm or 380 nm.

Data were searched from the database SeaBaSS. A first screening of the data identified many noisy and biased spectra. As a first baseline correction, the residual NIR value, which was estimated as the average $a_{ph}$ between 780 nm and 800 nm, was subtracted. Then, a PANGAEA search was performed. It delivered many excellent spectra, collected in seven Polastern cruises (Soppa et al., 2013b;Liu et al., 2019b, c;Bracher, 2019;Bracher et al., 2021j;Bracher et al., 2021d;Bracher and Taylor, 2021), one Sonne cruise (Bracher et al., 2021l) and one Heincke cruise (Bracher et al., 2021e). The PACE dataset (Casey et al., 2020) was also used, in particular by the PI Schaeffer and from the Biosope cruise. Then, to increase the high end of the range, Castagna's dataset on Belgian coastal and inland waters (Castagna et al., 2022) was used. Their published $a_{ph}$ was only available until 380 nm, so A. Castagna kindly made available $a_{ph}$ spectra especially processed for this investigation down to 350 nm, though expressing some methodological concerns about the data accuracy in the UV. Finally, a new CNR small dataset from a recent cruise (publication in preparation) has also been included in the global dataset.

Data quality among databases varied greatly, from generally poor within SeaBaSS to the carefully produced Castagna's spectra. Given the high amount of data in total, it was preferred to apply rather aggressive filter selection criteria. Spectra were smoothed with an 11 nm rectangular moving window to eliminate random noise introduced by the spectro-photometers. A relative noise parameter was calculated as the standard deviation of the difference between the unfiltered and the filtered $a_{ph}$, divided by a guess of the chlorophyll concentration based on $a_{ph}(665)$ (details below). Spectra were retained if this noise parameter was lower than 0.002, except for the Biosope dataset, where the threshold was raised at 0.004 in order to keep some low-end $a_{ph}$ that were very necessary for their representativity of the lowest $a_{ph}$ in the world. Additionally, the absolute value of the second derivative with respect to the wavelength $|a''_{ph}|$ was calculated as a measure of spectral noise. The 90$^{th}$ percentile of $|a''_{ph}|$ between 350 nm and 800 nm was stored. Only spectra having this percentile lower than 0.0032 were selected.

Further exclusion criteria were applied based on the spectral shapes. We defined the following indexes:

$$
\begin{aligned}
m_{UV} &= \min\{a_{ph}(\lambda \in [350\text{ nm}, 450\text{ nm}])\} \\
M_{UV} &= \max\{a_{ph}(\lambda \in [350\text{ nm}, 450\text{ nm}])\} \\
M_{G} &= \max\{a_{ph}(\lambda \in [550\text{ nm}, 560\text{ nm}])\} \\
I_{CHL} &= \max\{a_{ph}(\lambda \in [650\text{ nm}, 700\text{ nm}])\} - \min\{a_{ph}(\lambda \in [650\text{ nm}, 700\text{ nm}])\}
\end{aligned}
\quad (1)
$$

Therefore, the following selection thresholds were applied, that were chosen based on experience so that clearly anomalous spectra would be discarded yet trying not to penalize natural variability. These were $m_{UV}/I_{CHL} > 0.1$, $M_{UV}/I_{CHL} < 6$ and $M_G/M_{CHL} < 2$. In particular, the thresholds involving the UV discarded many spectra that raised excessively, likely consequence of insufficient bleaching of the filtered sample, or that tended to zero or even negative values. At the green range, it was assumed that the spectrum shall present a valley or at least a value that is not much larger than the chlorophyll peak. Other than these thresholds, some spectra exhibited secondary peaks very distant from 676 nm, which was likely a sign of spectral misalignment. Therefore, it was required that such peak was between 670 nm and 681 nm for inclusion.

All the filtering procedures led to the selection of 3025 high quality $a_{ph}$ spectra, representing a very wide range of values and water types.

#### 2.1.2 CDOM absorption

CDOM absorption at 440 nm ($a_g(440)$ or $Y$) is one of the three independent variables of the bio-optical modelling. Its value is therefore given. The full spectrum is covered by assuming a spectral variation, modelled here as the usual exponential shape. The value of the spectral slope $S_g$ must be determined. For this sake, a pool of in situ CDOM absorption spectra were collected. CDOM is stored by filtering seawater with 0.2 µm pore size filters. Absorption is measured through light transmission, as the scattering of the sample can be considered negligible. The most common measurement instrument is a bench spectrophotometer, where water is poured in a cuvette of a given path length, usually between 1 cm and 10 cm. In clear waters, because of the short path length that makes resulting data very noisy, a liquid waveguide capillary cell (LWCC) system like UltraPath™ (World Precision Instruments, Inc.) is preferred. They allow much larger path lengths, up to 2 m, therefore obtaining proper optical densities for a given sample, even in the clearest waters. In this article, only open access CDOM data measured with UltraPath were selected in open ocean waters, whereas in complex coastal and inland waters, cuvette-based measurements were accepted as well. Therefore, the pooled CDOM data consists of the PACE datasets Schaeffer, Biosope and Mouw, Castagna's measurements, as well as a large PANGAEA dataset based on several Polarstern cruises (Bracher et al., 2021a;Bracher et al., 2021b;Bracher et al.,

2021i;Bracher et al., 2021h) and some smaller campaigns in coastal areas (Juhls et al., 2019;Hölemann et al., 2020;Bracher et al., 2021g;Pykäri, 2022). In all cases, data had to be provided at the range from 350 nm to 750 nm and close to the surface.

Spectra were fitted to a decreasing exponential function with a given offset, $\hat{a}_{g,mod} = a_g(\lambda_0)e^{-S_g(\lambda-\lambda_0)} + a_{g,off}$ using non-linear least squares, with a bi-square weighting function to minimize the effect of outliers. Then, the offset was removed: $a_{g,mod} = \hat{a}_{g,mod} - a_{g,off}$. Notably, fits were made in linear scale, as making them in logarithmic scale would artificially raise the weight of spectral regions where CDOM is less relevant. Fits were required that $r^2 > 0.995$, to exclude eventual anomalous shapes. In total, 1168 spectra were retained.

#### 2.1.3 NAP absorption

As with CDOM, NAP absorption spectra ($a_{NAP}$) are not introduced directly in the radiative transfer simulations but modelled as exponential functions. Data selection again prioritized high quality as the data quantity was sufficient to derive the statistical relationships. Here, a PANGAEA search delivered data from various Polarstern cruises (Gonçalves-Araujo et al., 2018;Liu et al., 2019a, d;Wiegmann et al., 2019;Bracher et al., 2021k;Bracher et al., 2021f, c;Bracher and Liu, 2021;Soppa et al., 2013b, a) and one Heincke cruise (Bracher et al., 2021c). From the PACE database, $a_{NAP}$ from the cruise Biosope and the PIs Mouw and Schaeffer was included. Castagna's measurements were also included, as well as recent CNR data.

As for CDOM, an exponential shape was fitted, $\hat{a}_{NAP,mod} = a_{NAP}(\lambda_0)e^{-S_{NAP}(\lambda-\lambda_0)} + a_{NAP,off}$ and then the offset was removed, $a_{NAP,mod} = \hat{a}_{NAP,mod} - a_{NAP,off}$. The condition $r^2 \geq 0.995$ was imposed, leading to a total of 1349 valid spectra.

#### 2.1.4 Particle backscattering

In situ particle backscattering $b_{bp}$ is not a Hydrolight input parameter in the configuration that was used, but it was needed for the determination of the bio-optical relationships of the particulate fraction, as it will be detailed below. In addition, it is desirable to collect a comprehensive dataset of $b_{bp}$ matched to

fractionated absorption components to check the consistency of crossed relationships in the synthetic dataset respect to those found in natural waters.

Specifically, $b_{bp}$ at 440 nm or near wavelength was searched. The availability of such data is very limited and the quality is essentially unknown. Here, a best effort exercise was made by collecting all available data, of an open source or not. These were found from NOMAD (Werdell and Bailey, 2005), the PACE datasets Biosope and from the PI Mouw (Casey et al., 2020), and from Castagna's dataset (Castagna et al., 2022). In this latter case, $b_{bp}$ was not available, but since such data are considering especially important for their very high values, $b_{bp}$ was inferred through semi-analytic closure from absorption and $R_{rs}$ (Lee et al., 2011). Finally, data collected in Australian waters by CSIRO (Blondeau-Patissier et al., 2009;Blondeau-Patissier et al., 2017;Cherukuru et al., 2016;Oubelkheir et al., 2023;Brando et al., 2012) was also included here. CSIRO's dataset contains several IOPs and OACs at reference wavelengths, that were used to develop some of the bio-optical relationships that were used to produce the synthetic dataset. In what regards backscattering specifically, $b_{bp}$ is only provided at the reference wavelength 555 nm and with an estimated spectral slope. For this specific dataset, the slope was not only used to shift $b_{bp}$ from 555 nm to 440 nm, but also for a part of the bio-optical modelling, detailed below.

### 2.2 Bio-optical modelling

Assuming an unpolarized submarine light field, IOPs consist of the absorption coefficient and the volume scattering function (VSF; symbol $\beta$), which can be broken down to the contribution of the single OAC:

$$\begin{cases} a(\lambda) = a_w(\lambda) + a_{ph}(\lambda) + a_{NAP}(\lambda) + a_g(\lambda) \\ \beta(\Psi,\lambda) = \beta_w(\Psi,\lambda) + \beta_{ph}(\Psi,\lambda) + \beta_{NAP}(\Psi,\lambda) \end{cases} \quad (2)$$

This formulation implies the assumption that dissolved material does not significantly scatter light in the optical domain.

Commonly, optical theory deals with angular integrals of the VSF. If it is integrated across all directions, one obtains the scattering coefficient, whereas if one integrates across the backward hemisphere, one obtains the backscattering coefficient:

$$\begin{cases} b(\lambda) = b_w(\lambda) + b_{ph}(\lambda) + b_{NAP}(\lambda) \\ b_b(\lambda) = b_{bw}(\lambda) + b_{b,ph}(\lambda) + b_{b,NAP}(\lambda) \end{cases} \quad (3)$$

Given a certain constituent, whether phytoplankton or NAP, its VSF is normalized by its scattering to obtain the phase function (PF):

$$\tilde{\beta}_x = \frac{\beta_x}{b_x}, x = ph \text{ or } NAP \quad (4)$$

For radiative transfer calculations, the PF must be a priori established for each OAC. The backscattering ratio is used to constraint the PF to a first order (Mobley et al., 2002):

$$\mathrm{B}_x = \frac{b_{b,x}}{b_x}, x = ph \text{ or } NAP \quad (5)$$

With this information, a phase function must be assigned to each type of particle, consistent with the given data, such as $\mathrm{B}_x$. It can be a measured phase function (He et al., 2017), but more commonly from a family of simulated functions after electromagnetic scattering calculations (Morel et al., 2002;Fournier and Forand, 1994)

Despite being the bio-optical modelling more accurate if the particulate material is decomposed into the phytoplanktonic and the non-phytoplanktonic parts (some discussion below), scattering meters do not measure the separate contributions of phytoplankton and NAP. Instead, their “particle” aggregates are measured:

$$\begin{cases} b_p(\lambda) = b_{ph}(\lambda) + b_{NAP}(\lambda) \\ b_{bp}(\lambda) = b_{b,ph}(\lambda) + b_{b,NAP}(\lambda) \end{cases} \quad (6)$$

This aggregation is therefore made when the bio-optical modelling of scattering is about to be evaluated against in situ data.

The various terms of the absorption and scattering budgets will be modelled as a function of the OACs as detailed in the next sections and summarized in Table 1.

**Table 1 Summary of the bio-optical modelling**

| | |
|---|---|
| $a_{ph}(\lambda)$ | $a_{ph}(\lambda)$ from a quality-controlled database, adjusted by a factor to verify $a_{ph}(670) = AC^E$, A=0.019093, E=0.95568 |
| $c_{ph}(\lambda)$<br>$\hat{\beta}_{ph}(\Psi)$ | $c_{ph}(\lambda) = c_{ph}(660)\left(\frac{660}{\lambda}\right)^{n_1}$ |

| | |
|---|---|
| | $n_1 = -0.4 + \frac{1.6 + 1.2\Re}{1 + C^{0.5}}$<br>$\Re \leftarrow U(0,1)$<br>$\hat{\beta}_{ph}(\Psi) \sim FF(B_{ph})$<br>$B_{ph} \leftarrow N(\mu, \sigma)$<br>$\mu = 0.002 + (0.01 - 0.002) \cdot \exp[-0.56 \log_{10}(C)]$<br>$\sigma = 0.001(3 - \log_{10}(C)) + 0.001$ |
| $a_{NAP}(\lambda)$ | $a_{NAP}(\lambda) = N a^*_{NAP}(440) \cdot e^{-S_{NAP}(\lambda - 440)}$<br>$\log_{10} a^*_{NAP}(440) \leftarrow N(\mu, \sigma)$<br>$\mu = a\, e^{\left(b \log_{10}\frac{C}{N} + c\right)}$<br>$a = -0.1886, b = -1.055, c = -1.27$<br>$\sigma = 0.2627$<br>$S_{NAP} \leftarrow \begin{cases} U(0.01,0.035) \text{ if } a_{NAP}(440) < 4 \cdot 10^{-4}\ \mathrm{m}^{-1} \\ \mathrm{Ln\ N}(-0.308x - 5.101, -0.0558x + 0.1164) \text{ if } a_{NAP}(440) \in [4 \cdot 10^{-4}, 0.06)\ \mathrm{m}^{-1} \\ N(0.011,0.016) \text{ if } a_{NAP}(440) \geq 0.06\ \mathrm{m}^{-1} \end{cases}$ |
| $c_{NAP}(\lambda)$, $\hat{\beta}_{NAP}(\Psi)$ | $c_{NAP}(\lambda) = c_{NAP}(440) \left(\frac{\lambda}{440}\right)^{-\gamma_{NAP}}$<br>$\gamma_{NAP} \leftarrow N(\mu, \sigma)$<br>$\mu = 0.7, \sigma = 0.3$<br>$c(440) = a_{NAP}(440) + b_{NAP}(440)$<br>$b_{NAP}(440) = \frac{b_{b,NAP}(440)}{B_{NAP}}$<br>$B_{NAP} \leftarrow U(0.01,0.02)$<br>$b_{b,NAP}(440) = T b_{bp}(440) - b_{ph}(440)$<br>$T = N + 0.07C$<br>$b^*_{bp}(440) = b^*_{bp}(555) \left(\frac{440}{555}\right)^{-\eta}$ |

| | |
|---|---|
| | $\eta \leftarrow Burr(\alpha, c, k)$<br>$\alpha = 0.854, c = 4.586, k = 1.108$<br>$\log_{10} b_{bp}^{*}(555) \leftarrow N(\mu, \sigma)$<br>$\mu = m \log_{10} a_{NAP}^{*}(440) + n$<br>$m = 0.6834, n = -0.9483$<br>$\sigma = 0.2627$<br>$\hat{\beta}_{NAP}(\Psi) \sim FF(B_{NAP})$ |
| $a_g(\lambda)$ | $a_g(\lambda) = Y e^{-S_g(\lambda - 440)}$<br>$S_g \leftarrow \begin{cases} U(0.01, 0.025) \text{ if } a_g(440) < 0.02 \text{ m}^{-1} \\ \text{N}(-0.00040161x + 0.017508, -0.0003012x + 0.001881) \text{ if } a_g(440) \in [0.02, 5) \text{ m}^{-1} \\ U(0.0143, 0.017) \text{ if } a_g(440) \geq 5 \text{ m}^{-1} \end{cases}$ |

**2.2.1 Optically active constituents**

It is set as a goal to generate a dataset that covers the widest possible range of optical water types. As such, the historic case 1 assumption is inappropriate, and an IOP definition based on a single index such as chlorophyll concentration is therefore not adopted. Instead, a generic three-variables model is used, in which variability is driven by: the chlorophyll concentration (C), the NAP concentration (N), and CDOM absorption at 440 nm (Y). However, if C, N and Y were completely independent, the bio-optical modelling would generate unrealistic IOP combinations. Instead, for a hypothetical large dataset that contains such variables, they are expected that they have a certain degree of general relationship, tighter for the smaller values, that are found in the ocean.

Here, the partial relationship between the three variables in logarithmic scale was modelled with the generation of 5000 triplets, following three Burr type XII random probability density functions, $x \leftarrow \text{B}(\alpha, \text{c}, \text{k})$, related by a cross correlation matrix among them with the off-diagonal elements $\rho_{CN} = 0.8$,

$\rho_{CY} = 0.75$, $\rho_{YN} = 0.6$. Then, the derived random numbers were transformed to the actual (C,N,Y), variables with $X = 10^{x-d}$, where $X$ is either C, N or Y, and $x$ is their logarithmic counterparts. This is summarized in Table 2. Finally, very few outliers $C > 1000\, mg\, m^{-3}, N > 2000\, g\, m^{-3}$ and $Y > 100\, m^{-1}$ (~0.2 % or less) were considered excessive and were re-generated with a log-normal distribution, with the mean and standard deviation calculated from the rest of the dataset.

**Table 2 Parameters of the probabilistic modelling of the optically active constituents**

| | Burr distribution parameters | | | Scale coefficient |
|---|---|---|---|---|
| Variable | α | c | K | d |
| Chlorophyll concentration (C) | 3 | 3 | 2 | 3 |
| Non-algal particles concentration (N) | 3 | 4 | 1 | 4 |
| CDOM absorption coefficient at 440 nm (Y) | 2 | 6 | 1.3 | 4 |

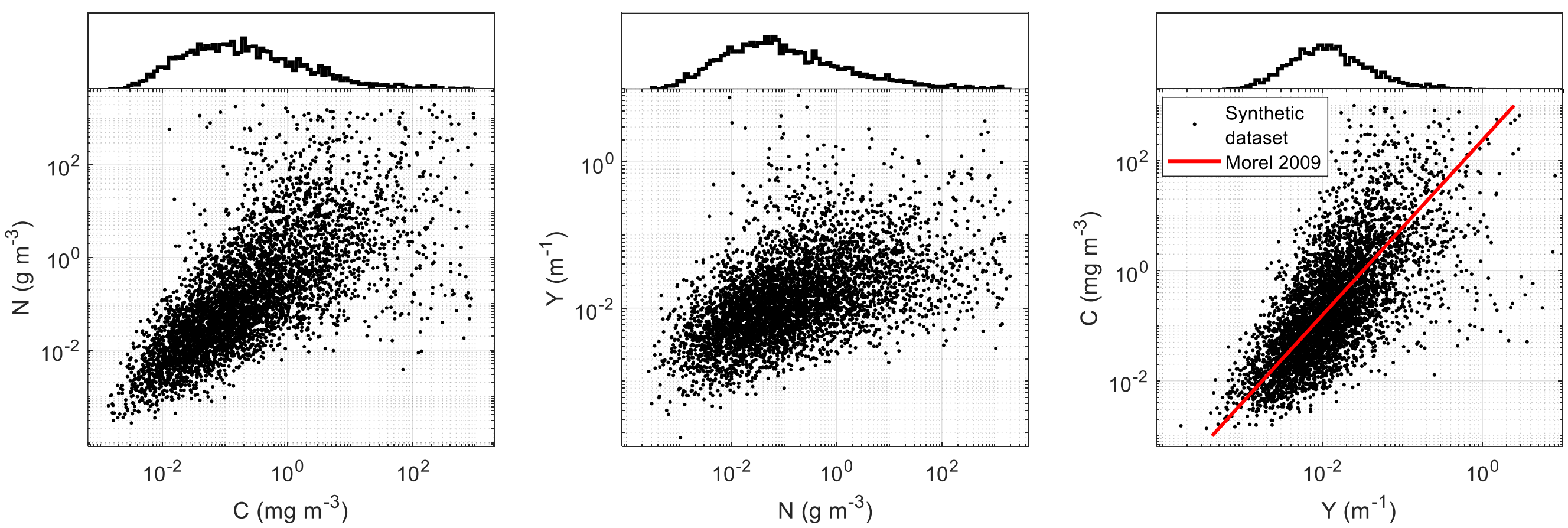

**Figure 1: Upper panels: histograms of the water constituents chlorophyll concentration (C), non-algal particles concentration (N) and CDOM absorption at 440 nm (Y). Lower panels: relationships between them. For the case between C and Y, the relationship by Morel (2009) in oceanic waters is added for comparison.**

In Fig. 1, the outcomes of OAC generation are depicted, showcasing a broad spectrum. The intentionally skewed distributions were formulated to mirror histograms observed in a comprehensive global dataset: frequencies of data surge from the lower values, peak at levels commonly encountered in global oceans, and gradually taper off at higher extremes. Concerning interrelationships, there is observable correlation,

with the empirical case 1 curve identified by Morel (2009) serving as a typical benchmark. However, as values ascend, the connection diminishes, consistent with expectations for coastal waters

#### 2.2.2 Phytoplankton absorption and scattering

Phytoplankton absorption $a_{ph}$ was modelled using data from the pool described in section 2.1.1. In order to generate phytoplankton diversity, it was important that, each time, a real $a_{ph}$ spectrum was used. A
395 similar approach to the $a_{ph}$ generation in the IOCCG dataset was followed, but first, it was found appropriate to revisit the relationship between C and $a_{ph}$. Matched data (Valente et al., 2022;Castagna et al., 2022) at several wavelengths (Fig. 2) revealed a tight linear relationship in log-log scale, though with some scatter, a part of which is attributable due to pigment variation. A power-law model was regressed at each wavelength:

$a_{ph}(\lambda) \approx A(\lambda)C^{E(\lambda)}$ (7)

**Table 3 Output variables of the regression between matched chlorophyll concentration and phytoplankton absorption of the merged datasets Valente et al. (2022) and Castagna et al. (2022) at several bands.**

| $\lambda\ (nm)$ | 411 | 443 | 489 | 510 | 555 | 670 |
|---|---|---|---|---|---|---|
| A | 0.043934 | 0.051348 | 0.03299 | 0.02132 | 0.0077002 | 0.019093 |
| E | 0.80289 | 0.77654 | 0.76732 | 0.8214 | 0.92914 | 0.95568 |
| n | 3509 | 3526 | 3525 | 3507 | 3231 | 2875 |
| RMSE (%) | 58.951 | 59.249 | 57.358 | 52.626 | 56.781 | 47.256 |
| $r^2$ | 0.85688 | 0.84553 | 0.84846 | 0.88033 | 0.89645 | 0.92553 |

Table 3 presents the regression outcomes, including the two model parameters (A,E), data number (n),
the root mean square in percent units and the coefficient of determination ($r^2$). A comparison to results from previous publications (Churilova et al., 2023;Bricaud et al., 1995) is made in Fig. 3.

Our results show that the 670 nm band has the highest capability for predicting C given $a_{ph}$. It is important to emphasize that this calculation does not model $a_{ph}$ for a specific C; rather, it associates each $a_{ph}$ with its characteristic "C", from inversion of eq. (7). This facilitates the sorting of the 3025 $a_{ph}$ spectra based
on "C", dividing them into 55 pools of specific "C" sub-ranges, each containing 55 spectra. Consequently,

for a given C value from the (C,N,Y) triplet, a random $a_{ph}$ spectrum from the corresponding pool is selected. Subsequently, the spectrum is adjusted by a factor so that $a_{ph}(670)$ equals the predicted $a_{ph}(670)$ from C, after eq. (7). This methodology guarantees consistency between $a_{ph}$ and empirical evidence for a given C while ensuring a broad diversity in $a_{ph}$ spectral shapes.

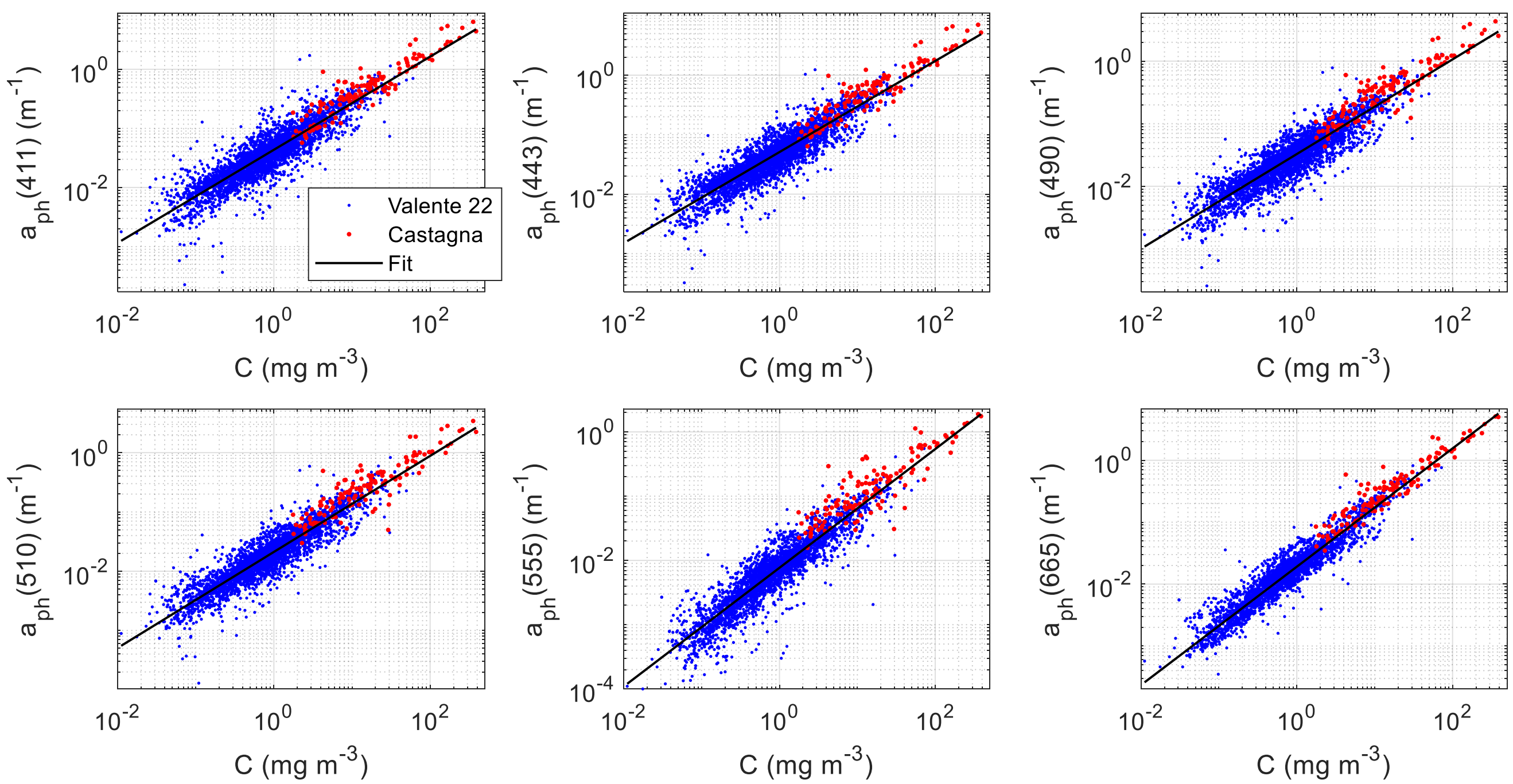

**Figure 2: Matched C and $a_{ph}$ data (Valente et al., 2022;Castagna et al., 2022) at six wavelengths. A linear fit in log-log form is displayed on top.**

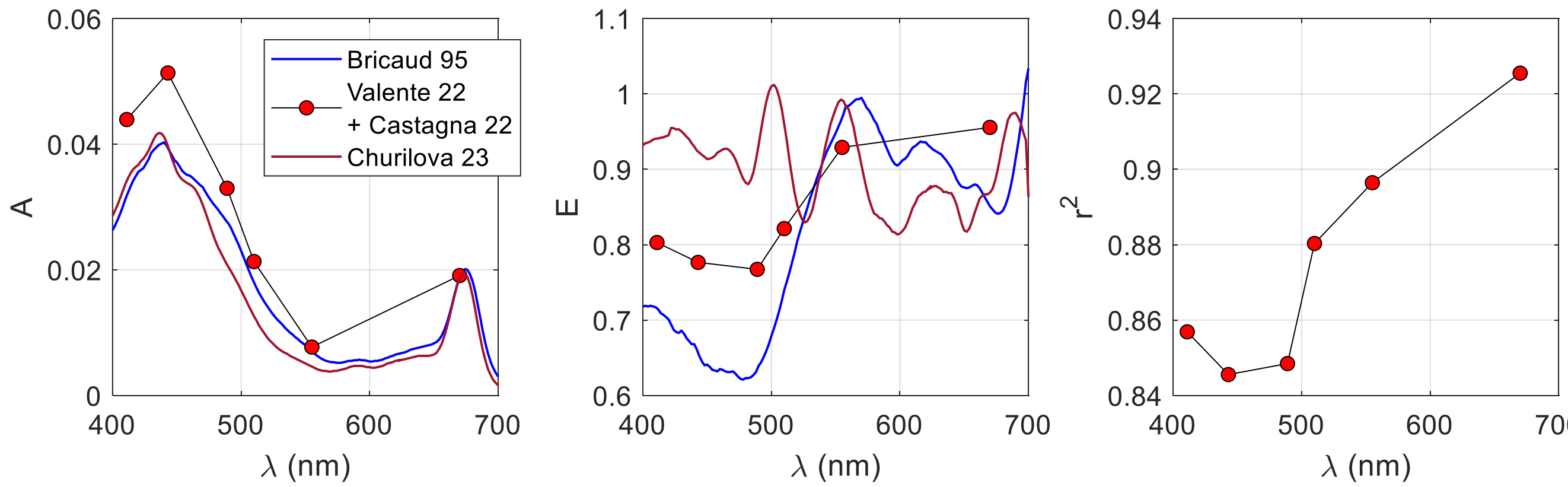

**Figure 3: Regression statistics the fit between C and $a_{ph}$ data of Fig. 2. Left and center plots are Bricaud's A and E parameters, whereas the right plot is the coefficient of determination ($r^2$).**

Phytoplankton scattering ($b_{ph}$) modelling unfortunately has much less background knowledge, mostly due to the lack of instruments that can measure in situ $b_{ph}$ or $b_{b,ph}$. Still, there are some notable modelling contributions based on electromagnetic theory (Lain et al., 2023;Poulin et al., 2018).

Upon this lack, it is often referred to historic measurements by Loisel and Morel (1998) of non-water beam attenuation coefficient at 660 nm $c_{nw}(660)$ with a transmissometer, matched to chlorophyll concentration in case 1 waters. For the surface layer, they found $c_{nw}(660) = 0.407C^{0.795}$. Furthermore, the authors reasonably assumed that the dissolved contribution was secondary, so $c_p(660) \approx c_{nw}(660)$. Unfortunately, this relationship was directly exported to phytoplankton scattering modelling used in the Coastcolour dataset, replacing $c_p(660)$ with $c_{ph}(660)$, ignoring that even in open sea waters, the non-algal scattering is considerable. Instead, the in the IOCCG dataset, a random coefficient was used for phytoplankton attenuation, while leaving the power dependence:

$c_{ph}(660) = p_3 C^h$ (8)

According to the IOCCG report, $h = 0.57$, although application of eq. (8) to the downloadable dataset reveals $h = 0.63$. Here, $h = 0.7$ is used.

On $p_3$, it was set random between 0.06 and 0.6 in the IOCCG dataset. Interestingly, that leaves the contribution of phytoplankton mostly below what found by Loisel and Morel (1998) for the total attenuation, which appears physically meaningful. The type of randomness of $p_3$ was not disclosed, but an inspection to the IOCCG dataset revealed that it was uniform. This parameter is left unchanged for the modelling of the current dataset given the absence of empirical evidence that justifies otherwise.

The spectral variation is set by assigning a power law to phytoplankton attenuation. This choice, i.e., modelling the spectral variation of attenuation rather than of scattering, with a simple and featureless function, has physical justification. A power law function provides a better fit for attenuation than for scattering, as the latter is affected by anomalous dispersion effects, that result in some negative peaks with the shape of an absorption spectrum, more evident at high phytoplankton concentrations (Bernard et al., 2009).

In the absence of further information, the same relationship as in the IOCCG dataset is used, that is:

$$c_{ph}(\lambda) = c_{ph}(660)\left(\frac{660}{\lambda}\right)^{n_1}, \text{with } n_1 = -0.4 + \frac{1.6+1.2\mathfrak{R}}{1+C^{0.5}} \quad (9)$$

With $\mathfrak{R}$ being a random number that follows a uniform distribution in the interval [0,1].

Given the randomness of $a_{ph}$ and $c_{ph}$, it is possible that some realizations generate cases where $a_{ph} \leq c_{ph}$, which is unphysical. Indeed, a given $a_{ph}$ represents a certain community assemblage, which have their specific scattering characteristics. Unfortunately, there is a lack of knowledge on how to parameterize scattering when absorption is known. There are some simplified modelling results using electromagnetic theory for certain phytoplankton species (Lain et al., 2023), although a general modelling theory of phytoplankton scattering linked to absorption is still non-existent. Thus, in this dataset, as in the precedent ones, of $a_{ph}$ and $c_{ph}$ are modelled independently. A condition was set, that if there were any bands at which $a_{ph} \leq c_{ph}$, the procedure for determining $a_{ph}$ and $c_{ph}$ should be repeated.

The remaining parameter that must be set to run Hydrolight is the phytoplankton backscattering ratio, $B_{ph} = \frac{b_{b,ph}}{b_{ph}}$. This parameter has not been given much importance in previous research, as it was considered relatively unimportant, so it is common to find it set to a constant value in the order of 0.006 or 0.01. It is indeed secondary in semi-analytical models that model $R_{rs}$ from $\frac{b_b}{a+b_b}$ or variations, but in bio-optical modelling it can be very relevant if $b_{ph}$ is fixed first, because then, $b_{b,ph}$ is implicitly determined through the choice of the respective phase function given $B_{ph}$ (Mobley et al., 2002), thereby setting the intensity of the signal.

We pursued a determination of $B_{ph}$ that was consistent with the general trend that phytoplankton size increases with C. This has a diminishing effect on $B_{ph}$ because larger $B_{ph}$ is associated with smaller particles, which scatter relatively more in the backward hemisphere respect to larger ones. Also, smaller particles have a larger surface area per unit volume, which enhances scattering. A single variable mechanistic model for $b_{bp}$ that agrees with this principle was presented in Brewin et al. (2012). In terms of the backscattering ratio, Twardowski et al. (2001; Fig. 11) presented pioneering results, for $B_p$ in their case. Here, to mimic such effect, we set

$$B_{ph} \sim N(\mu, \sigma)$$

$$\mu = 0.002 + (0.01 - 0.002)\exp[-0.56\log_{10}(C)], \sigma = 0.001(3 - \log_{10}(C)) + 0.001 \quad (10)$$

To avoid unlikely low $B_{ph}$ values after eq. (10), any realization delivering $B_{ph}$<0.001 was set to 0.001 as a lower limit.

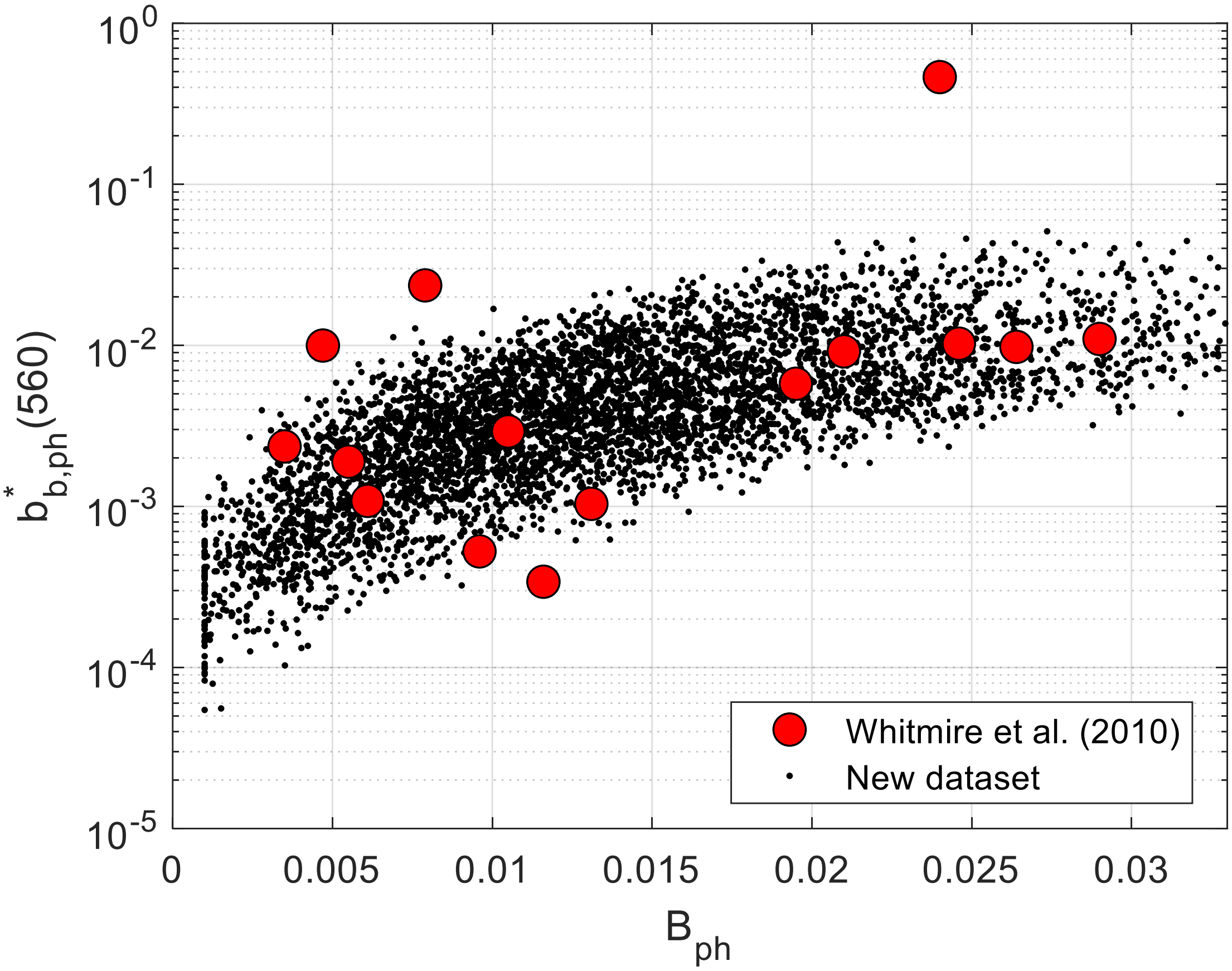

**Figure 4: Phytoplankton backscattering ratio $B_{ph}$ vs. phytoplankton specific backscattering coefficient at 560 nm $b^*_{b,ph}(560)$. Black dots: synthetic dataset. Red dots: data in Whitmire et al. (2010)**

The modelling of phytoplankton scattering we just presented has some possible independent validation. Whitmire et al. (2010) presented unique data of chlorophyll concentration matched to scattering and backscattering for an array of phytoplankton cultures. Their data of the chlorophyll-specific phytoplankton backscattering coefficient, i.e., $b^*_{b,ph} = \frac{b_{b,ph}}{C}$ at 560 nm, matched to $B_{ph}$ are fairly well on top of the data cloud of this synthetic dataset (Fig. 4), also verifying the positive correlation of the two variables to a first order.

#### 2.2.3 NAP absorption and scattering

Bio-optical modelling of NAP absorption $a_{NAP}$ requires linking it to the mass concentration, N, through the specific absorption (to NAP concentration) $a_{NAP}^{*}$. Results in Blondeau-Patissier et al. (2009) suggested that $a_{NAP}^{*}(440)$ varies between 0.001 and 0.1 m$^{2}$ g$^{-1}$. One may assume that such value depends on the type of particles. Following this consideration, here, the ratio C/N is proposed as a first-order predictor of $a_{NAP}^{*}(440)$. This dependence assumes that non-algal particles absorb more efficiently in the relatively higher presence of chlorophyll, which suggests that they may be of biogenic origin to a larger extent than if the chlorophyll concentration was relatively lower, where they may be more of a mineral origin instead. CSIRO data confirmed some degree of covariation (Fig. 5). The fit to the CSIRO data was made in logarithmic scale, so $y = \log_{10}[a_{NAP}^{*}(440)]$ was regressed as a function of $x = \log_{10}\left(\frac{C}{N}\right)$, proposing a functional form of the type $y = a\exp(bx) + c$. A robust regression (bi-square weighting) gave $a = -0.1886, b = -1.0551, c = -1.2700$. The standard deviation of the fit was $\sigma = 0.2627$. To generate the synthetic data, given C/N, the regression curve was applied and then a random value, generated with a normal distribution $N(0, \sigma)$ was added, in order to replicate the spread found in real data. $\frac{C}{N}$ in our synthetic dataset covers a wider range than CSIRO's data, so, to avoid producing resulting synthetic $a_{NAP}^{*}(440)$ values much out of the range of the measured data, the lower and upper bounds of -3 and -0.5 were set for $\log_{10}[a_{NAP}^{*}(440)]$. The results are shown in Fig. 5.

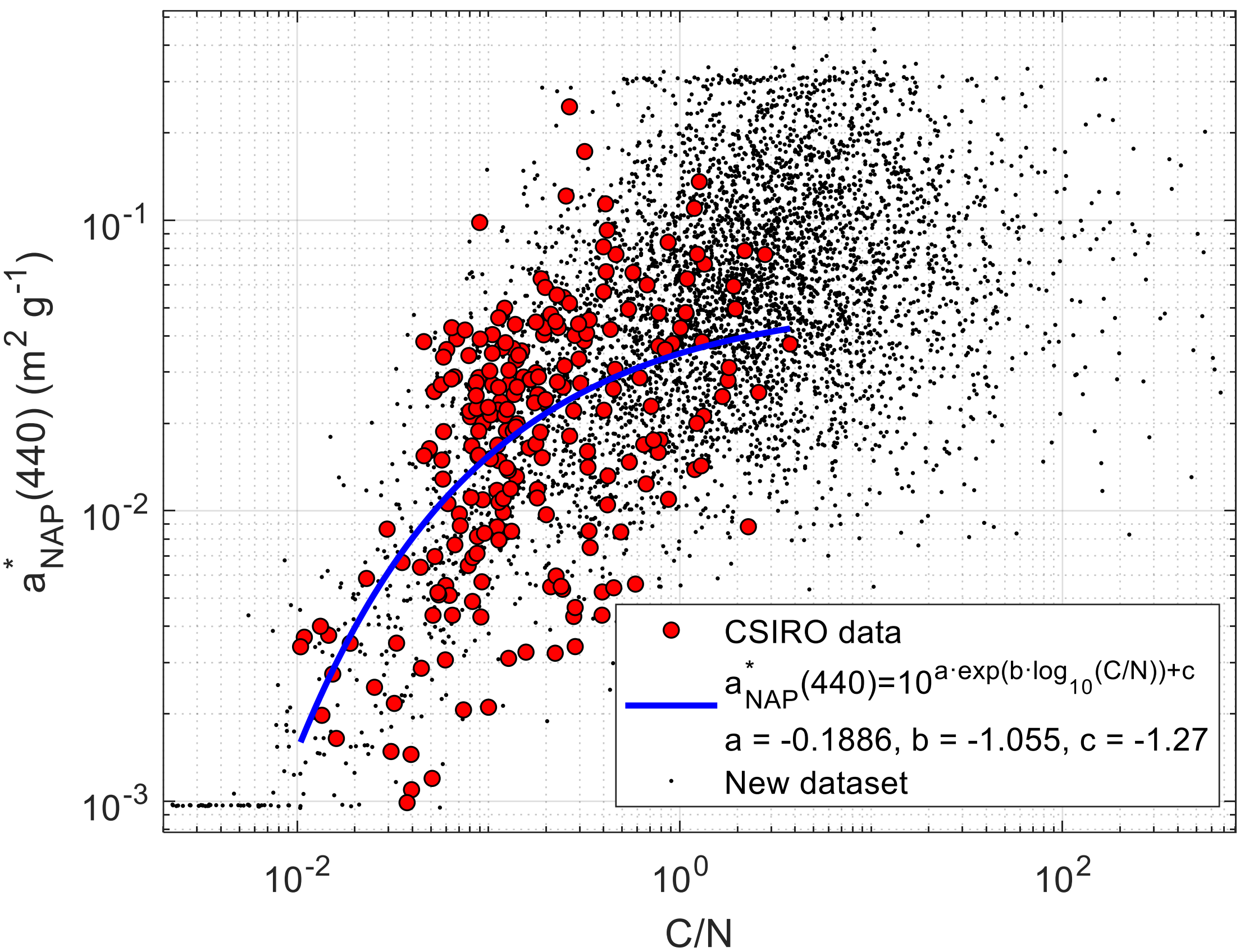

**Figure 5: Non-algal particles specific absorption coefficient at 440 nm $a^*_{NAP}(440)$, plotted as a function of the chlorophyll to NAP concentrations ratio C/N. Results for CSIRO data in red dots, a best fit in blue, and generated data for the synthetic dataset (black dots).**

Posteriorly, it is necessary to project $a^*_{NAP}(440)$ to all bands. It can be done by assuming an exponential spectral shape and then guessing a spectral slope ($S_{NAP}$). Historic data suggested an average $S_{NAP}$ value of 0.0123 nm$^{-1}$ (Babin et al., 2003), though with a visible spread. Using a single average $S_{NAP}$ for all simulations removes optical diversity and likely generates $a^*_{NAP}$ spectra that are unlikely for some regions. It is a better choice to generate a prediction function for $S_{NAP}$ given the available information. After the exponential fits for each of the compiled $a_{NAP}$ spectra, detailed in section 2.1.3, $a_{NAP}(440)$ and $S_{NAP}$ were calculated and plotted together in Fig. 6.

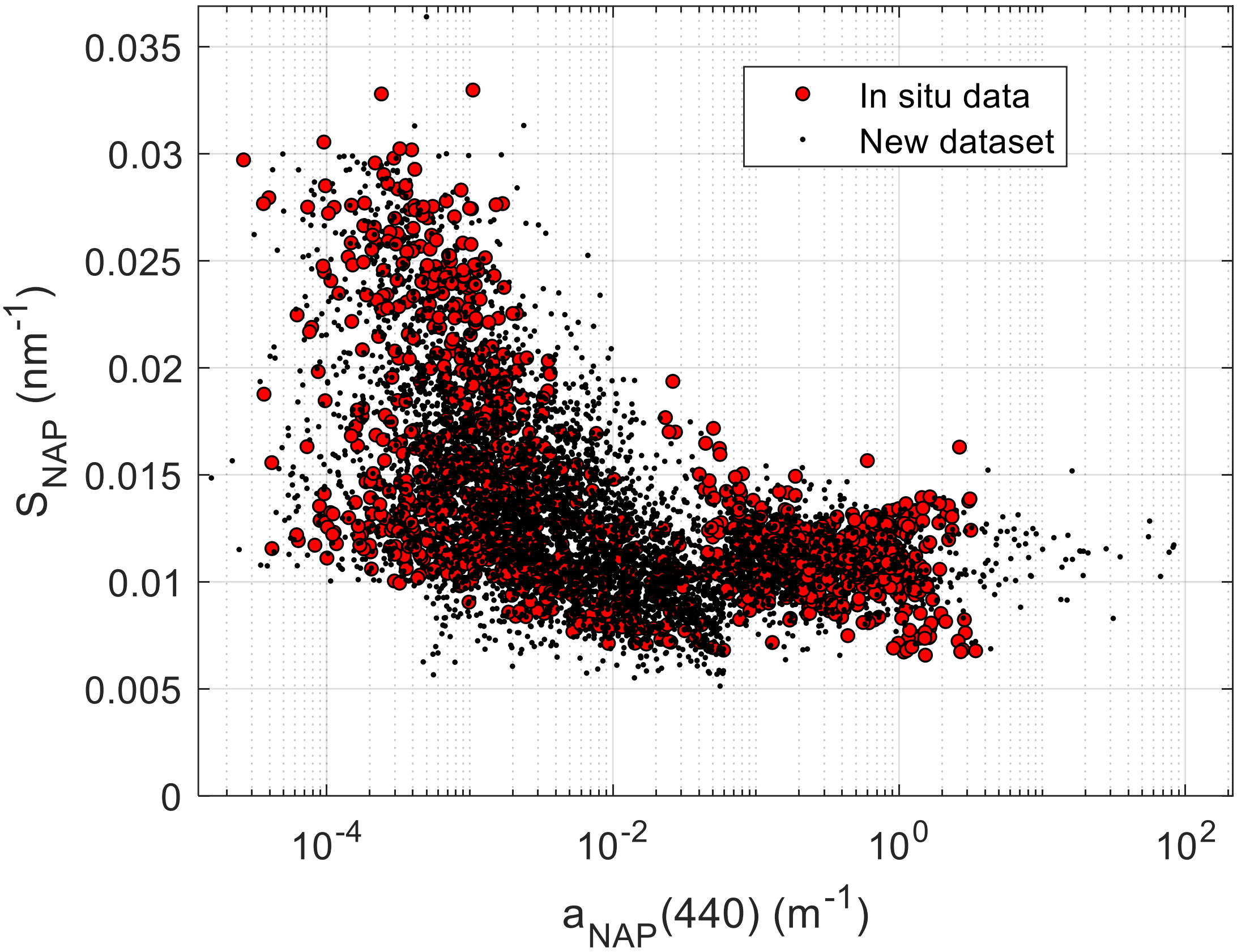

**Figure 6: Non-algal particles absorption spectral slope ($S_{NAP}$), plotted as a function of the NAP absorption coefficient at 440 nm ($a_{NAP}(440)$). Red dots: in situ data. Black dots: synthetic data.**

The data distribution in Fig. 6 shows a $S_{NAP}$ spread that largely varies depending on the range. For very small values, data shows no particular pattern between two bounds, so a uniform distribution was found adequate. For the middle range, the distribution somewhat narrows as $a_{NAP}(440)$ increases, and data
shows some positive skewness, which is well represented by a log-normal curve. For the higher range, a gaussian distribution was observed, in agreement with Babin et al. (2003). Therefore, given $x = \log_{10}[a_{NAP}(440)]$, $S_{NAP}$ was modelled as a piece-wise random distribution:

$$S_{NAP} \leftarrow \begin{cases} U(0.01, 0.035) \text{ if } a_{NAP}(440) < 4 \cdot 10^{-4}\ \mathrm{m}^{-1} \\ \mathrm{Ln\ N}(-0.308x - 5.101, -0.0558x + 0.1164) \text{ if } a_{NAP}(440) \in [4 \cdot 10^{-4}, 0.06)\ \mathrm{m}^{-1} \\ N(0.011, 0.016) \text{ if } a_{NAP}(440) \geq 0.06\ \mathrm{m}^{-1} \end{cases} \quad (11)$$

Where $U(a,b)$, $Ln\,N(\mu,\sigma)$ and $N\sigma$ are the uniform, log-normal and normal distributions, respectively. The random parameterization for $S_{NAP}$ in eq. (11) is rather convoluted. However, it ensures fitness to a high quality and large in situ dataset present in Fig. 6, and it does not generate outliers.

NAP scattering needs bio-optical modelling too. Approaches that model NAP absorption and scattering independently may generate unrealistic IOPs for that particular material. It is beneficial to look for relationships that link NAP scattering to NAP absorption, as it is expected to occur in reality.

The CSIRO dataset provides $b_{bp}^{*}(555)$ data, concurrent to $a_{NAP}^{*}(440)$. It must be clarified that $b_{bp}^{*}$ has been defined by normalizing $b_{bp}$ to the total suspended matter concentration (T), not to be confused with non-algal particles concentration N, as the latter is only a fraction of the former, which also contains the phytoplanktonic part. Brando and Dekker (2003) proposed a somewhat crude relationship, $T = N + 0.07C$, where both T and N are expressed in the usual units of g m$^{-3}$ and C is in mg m$^{-3}$. For interested readers, such relationship was derived from measurements at a shallow, turbid eutrophic lake in The Netherlands (Gons et al., 1992).

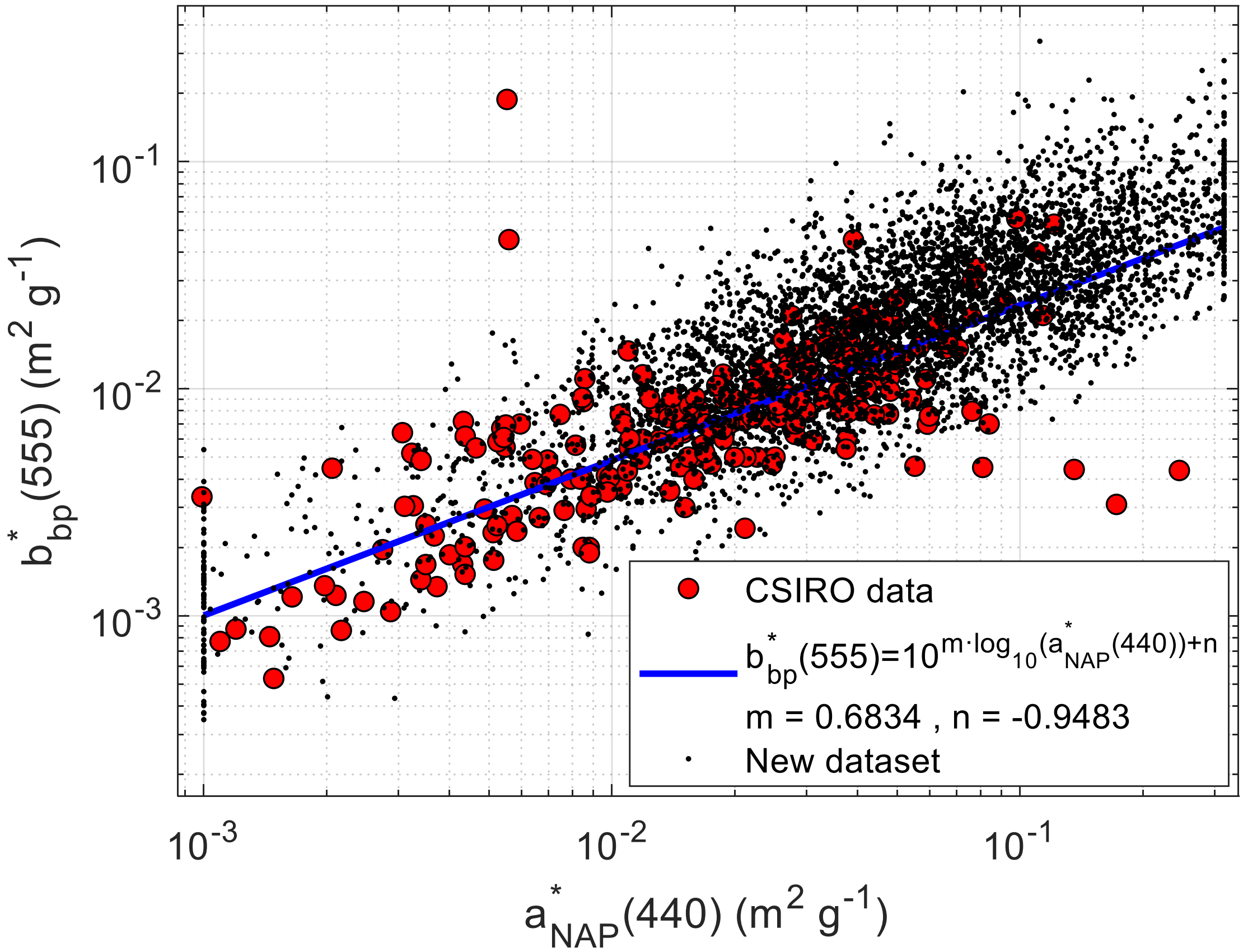

**Figure 7: Specific particle backscattering coefficient at 555 nm $b^*_{bp}(555)$, plotted as a function of the non-algal particles specific absorption coefficient at 440 nm $a^*_{NAP}(440)$. Results for CSIRO data in red dots, a best fit in blue, and generated data for the synthetic dataset (black dots).**

The relationship between $a^*_{NAP}(440)$ and $b^*_{bp}(555)$ data appeared to be very significant (Fig. 7, red dots). A linear trend was a very good fit between the log-transformed variables, with a slope m=0.6834 and an intercept n=-0.9483. The data spread followed a normal distribution ($\sigma = 0.2627$) after removing the trend line. To reproduce this spread in the synthetic dataset, a random number following a random normal distribution $N(0, \sigma)$ was added to the fit-predicted $b^*_{bp}(555)$, prior to conversion to linear scale again. Results of the generated data cloud are seen in Fig. 7, black dots.

Completing the bio-optical modelling for NAP requires that $b^*_{bp}$ is given at 440 nm, which implies projecting $b^*_{bp}$ from 555 nm to 440 nm with some sort of spectral parameter. CSIRO data provides an estimate of the particle backscattering spectral slope (η) for every data point. For the synthetic data

generation, a modelling function for η must be derived. No relationship between η and any other parameter within the CSIRO dataset was found, so its histogram was fitted to a random Burr distribution with the parameters $\alpha = 0.854, c = 4.586, k = 1.108$, shown in Fig. 8.

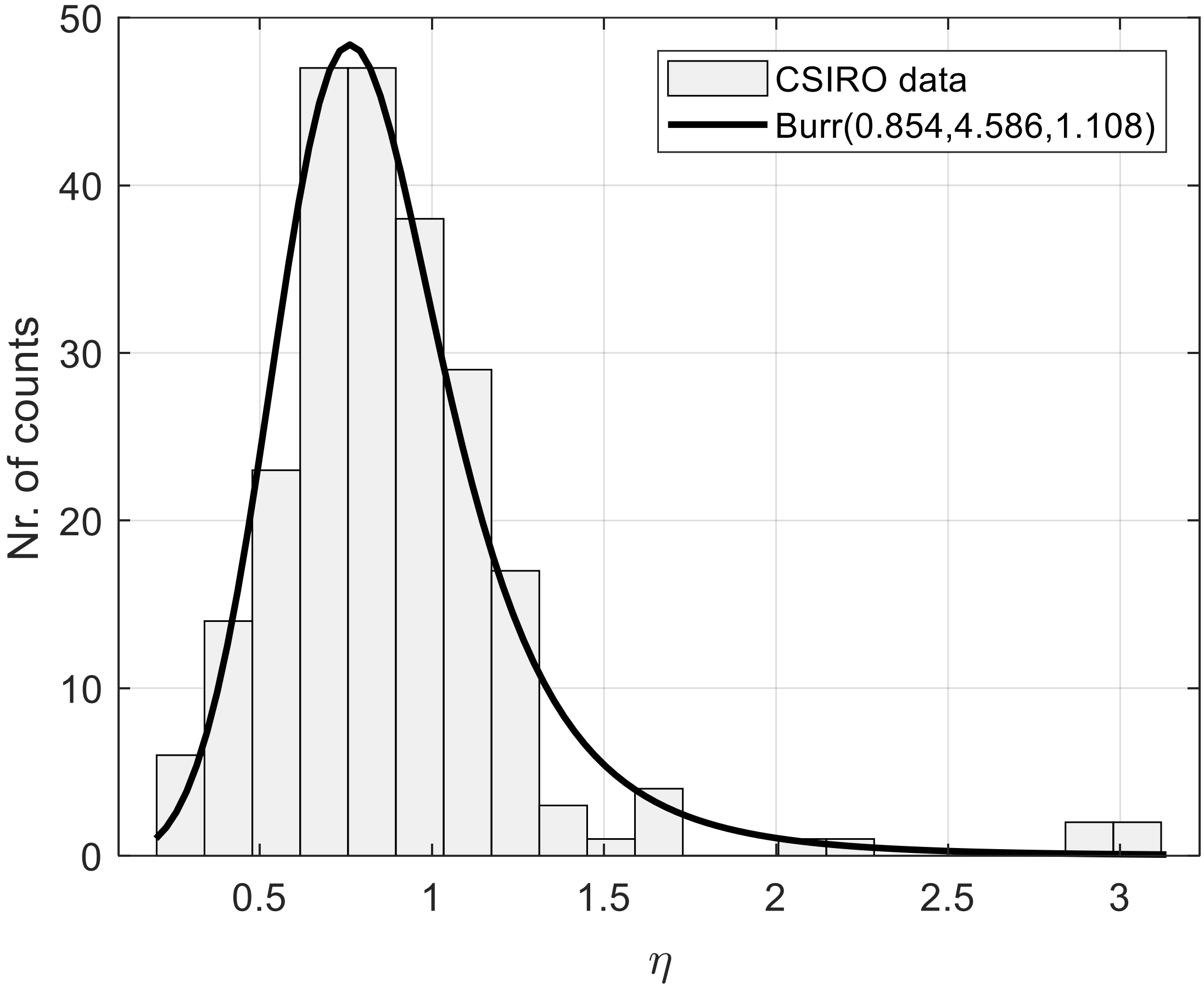

**Figure 8: Histogram of the particle backscattering coefficient spectral slope (η). A Burr Type XII fitted distribution is plotted on top.**

Therefore, the slope $\eta$ was randomly generated with such distribution, so that $b_{bp}^*(440) = b_{bp}^*(555)\left(\frac{440}{555}\right)^{-\eta}$. It must be noted that this slope is only used in this step and it is not used to propagate backscattering or any other IOP to the full spectral range. In the bio-optical modelling of NAP, as well as of phytoplankton, a spectral shape is assumed for attenuation, not backscattering.

The NAP backscattering is derived by subtraction of the phytoplanktonic part, which is known from section 2.2.2:

$b_{b,NAP}(440) = b^{*}_{bp}(440) \cdot T - b_{b,ph}(440)$ (12)

A backscattering ratio for NAP ($B_{NAP}$) must be assumed to obtain $b_{NAP}(440)$ and $c_{NAP}(440)$. There are no direct measurements of $B_{NAP}$ given the current impossibility of measuring NAP scattering parameters in the field. Nevertheless, this poses a minor problem for radiative transfer calculations, especially for remote sensing applications. As long as $b_{b,NAP}$ is fixed, $B_{NAP}$ is relatively unimportant, as one can deduct from simplified analytical models for reflectance or diffuse attenuation. If $b_{NAP}$ were fixed instead, $B_{NAP}$ would be a fundamental parameter, as it would implicitly set $b_{b,NAP}$, in a much less accurate fashion. $B_{NAP}$ is here fixed as a random number, following a uniform distribution between 0.01 and 0.02:

$B_{NAP} \leftarrow U(0.01,0.02)$ (13)

Therefore:

$b_{NAP}(440) = \frac{b_{b,NAP}(440)}{B_{NAP}}$ (14)

Then, the attenuation at 440 nm is expressed as a function of values that are all known:

$c_{NAP}(440) = a^{*}_{NAP}(440) \cdot N + b_{NAP}(440)$ (15)

NAP attenuation is extended to all wavelengths as a power law. As for phytoplankton, it is preferred to fit a power law to attenuation than to scattering, though recognizing that, given the much featureless shapes of NAP absorption, a fit to scattering may be realistic too. A $c_{NAP}$ spectral slope $\gamma_{NAP}$ must be derived. This parameter is largely unknown as it cannot be measured in the field. Here, an educated guess is made, generating $\gamma_{NAP}$ randomly, with $\gamma_{NAP} \leftarrow N(0.7,0.3)$.

#### 2.2.4 CDOM absorption

The same procedure as for the NAP absorption coefficient is followed here, as detailed in section 2.1.2: exponential functions were fitted to the $a_g$ spectra, and from those regressions having very high correlation, $a_g(440)$ and $S_g$ were retained (Fig. 9). The middle section shows a data spread, whose mean and standard deviation decrease with $a_g(440)$. Variation in the lower and upper range ends could not be

linked to any parameter, so that $S_g$ was modelled as uniform distributions, fairly within the data range. Overall, $S_g$ was then modelled as a piece-wise random distribution. Given $x = \log_{10}[a_g(440)]$:

$$S_g \leftarrow \begin{cases} U(0.01,0.025) \text{ if } a_g(440) < 0.02 \text{ m}^{-1} \\ \mathrm{N}(-0.00040161x + 0.017508, -0.0003012x + 0.001881) \text{ if } a_g(440) \in [0.02,5) \text{ m}^{-1} \\ U(0.0143,0.017) \text{ if } a_g(440) \geq 5 \text{ m}^{-1} \end{cases} \quad (16)$$

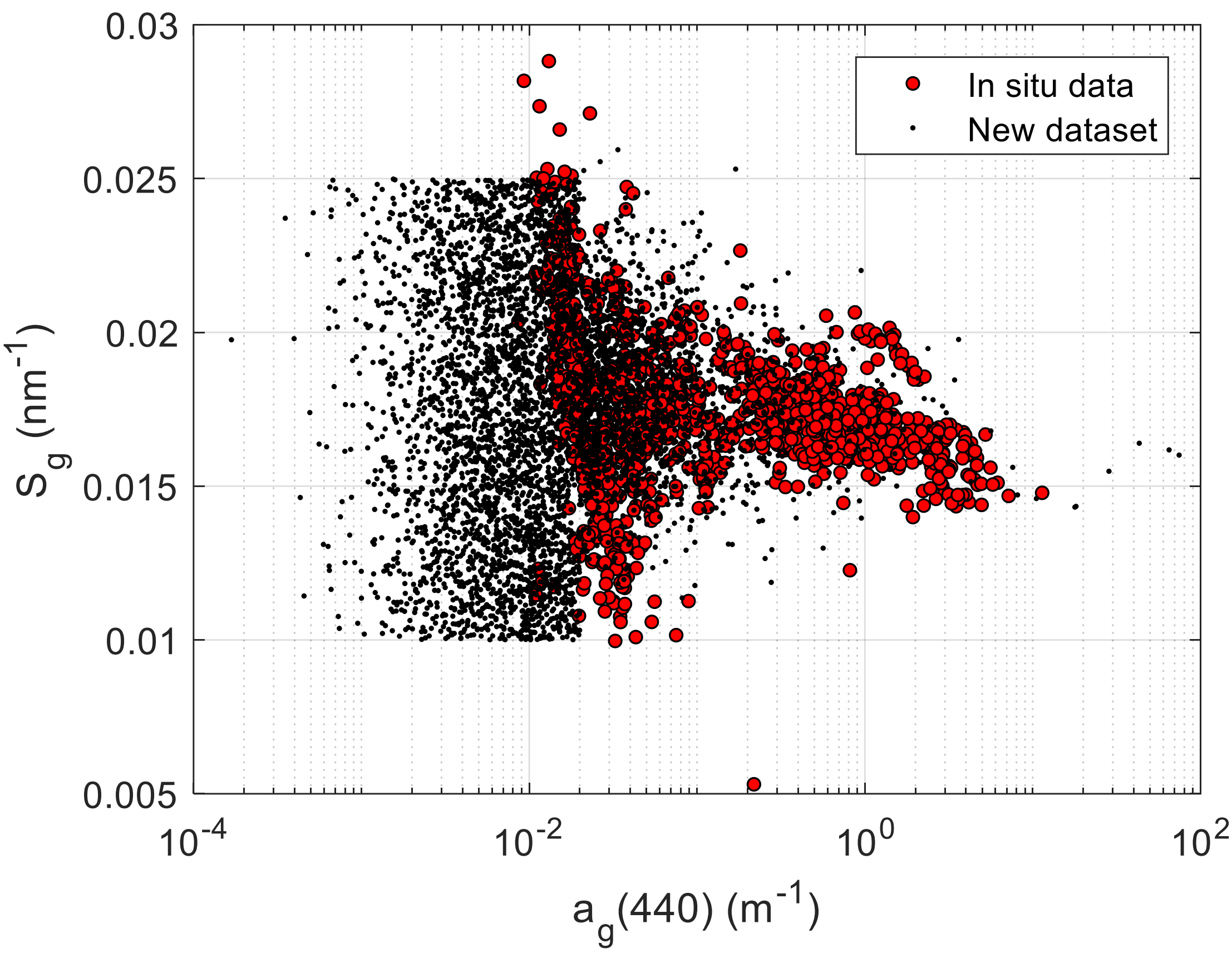

**Figure 9: CDOM spectral slope ($S_g$), plotted as a function of the CDOM absorption coefficient at 440 nm ($a_g(440)$). Red dots: in situ data. Black dots: synthetic data.**

It is shown that the synthetic dataset includes many points below the lower $a_g(440)$ limit. This is a consequence of the well-known under-sampling of the oligotrophic oceans. Extrapolation may raise some concerns, but on one hand, $S_g$ values are well bounded in this part of the range, and on the other hand,

one must also note that $a_g(440)$ becomes very low, so that potential errors in $S_g$ are not relevant for the absorption budget.

#### 2.2.5 Pure water absorption and scattering

Pure liquid water absorbs electromagnetic radiation, which can be intuitively explained as the energy being used by the two O-H molecular bonds to vibrate at given resonant frequencies, creating an absorption spectrum with characteristic maxima and minima at specific wavelengths.

Literature offers partial spectral range measurements for pure water absorption, owing to the specific requirements and challenges inherent in such measurements across different spectral regions. Factors such as signal-to-noise ratio, sample purity, and instrument cleanliness contribute to this variability. When compiling a comprehensive dataset spanning a broad range, a crucial step involves normalization to a common temperature. Fortunately, this merging process was already undertaken within the framework of an ESA project (Roettgers et al., 2016), where the "water optical properties processor" produced a consolidated dataset of pure water absorption, normalized to 20°C. Notably, this dataset encompasses measurements by Mason et al. (2016) from UV to green wavelengths, revealing lower water absorption in the UV and blue regions than previously documented, thanks to meticulous sample preparation and precise measurements. In other spectral regions, data from various authors are merged, sometimes overlapping spectrally and sometimes not. Overall, the WOPP pure water absorption data can be considered the state of the art. For comprehensive insights, readers are directed to the project report.

When marine salts are dissolved, the ions are dissociated and create a stable solution whose absorption can be related to that of pure water proportionally to the concentration of salt for the range of salinities that is commonly encountered, although this proportionality coefficient is wavelength dependent. Temperature affects absorption in a similar manner, thus leading to:

$$a_w(T,S) = a_w(T_0,0) + \Psi_T(T - T_0) + \Psi_S S \quad (17)$$

To the WOPP pure water merged absorption, a shift to an average ocean salinity of S=35 PSU was made with eq. (17), using the $\Psi_S$ coefficient provided by Roettgers et al. (2014) for artificial seawater.

Scattering by pure water finds explanation with the Smoluchowski-Einstein fluctuation theory of light scattering (Zhang and Hu, 2021), according to which, a certain volume of water can be seen as made of

smaller sub-volumes that contain, on average, the same number of water molecules. However, the instantaneous numbers vary among them due to random thermal motions at the molecular level, resulting in microscopic density fluctuations that induce scattering. In the presence of solutes such as salts, this effect is magnified, as fluctuations in the spatial arrangement of dissolved ions lead to variations in the overall refractive index. For common ocean salinities, scattering is augmented by approximately 30% respect to fresh water. Recent work by Zhang and Hu (2021) provides a comprehensive review of this theory, offering the most precise estimates to date (likely within ±2-4%). Nevertheless, rigorous experimental validation remains imperative. The formulas provided as supplementary material in their paper were employed to compute seawater scattering, assuming a temperature of T=20°C and a salinity of S=35 PSU, as for the absorption data.

## 3. Results of the synthetic dataset

### 3.1 Modelled IOPs

The bio-optical modelling detailed in the section 2.2 generated the IOPs that determine the resulting light field and related AOPs, given the boundary conditions. These bio-optical relationships have been individually assessed and consistency with literature and with new data has been ensured. However, a further test is desirable, that implies checking the crossed relationship between different commonly measured IOPs, compared to all available in situ data, in order to verify that the relationships that are found in the world's waters are represented.

The following publicly available in situ data were used: BIOSOPE cruise data from the clearest ultraoligotrophic waters of the south-Pacific gyre, plus some stations in coastal upwelling water off Perú (Casey et al., 2020), the NOMAD dataset (Werdell and Bailey, 2005), Castagna's data in Belgian coastal and inland waters (Castagna et al., 2022), measurements in coastal European waters (Massicotte et al., 2023), Mouw's data in Lake Superior (Casey et al., 2020), and recent measurements in Svalvard (Petit et al., 2022). In addition, two datasets not yet publicly available were queried to the authors, who kindly sent them for use in this article: data from the Persian Gulf (Moradi and Arabi, 2023) and from Australian waters (Blondeau-Patissier et al., 2009;Blondeau-Patissier et al., 2017;Cherukuru et al., 2016;Oubelkheir et al., 2023;Brando et al., 2012).

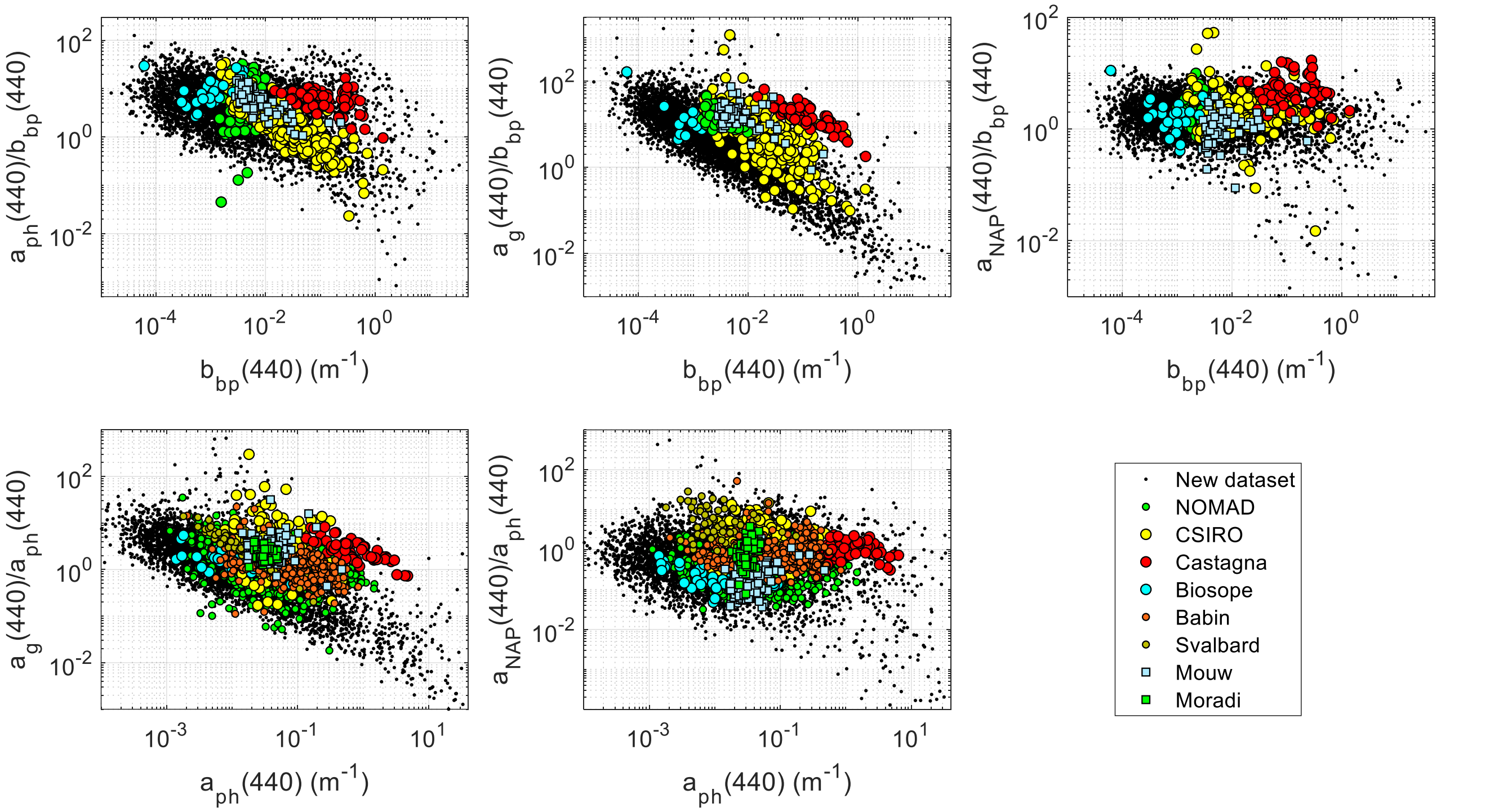

**Figure 10: IOP cross-relationship comparison between the synthetic dataset and various in situ datasets.**

Fig. 10 presents relationships among various IOPs at the reference wavelength 440 nm. The upper panels study the three non-water absorption components with respect to particle backscattering and the two lower panels study the CDOM and NAP absorption with respect to phytoplankton absorption. Given that any pair of IOPs are expected to linearly covary to the first degree, the vertical axis plots the ratio between the two, so that the linear covariation is eliminated, restricting the dynamic range and highlighting the differences among datasets. The plots show that available measurements in different regions and seasons cover different regions of the data space, and that the synthetic dataset nicely encompasses all of them. The plots show also areas where the synthetic dataset does not have correspondence to in situ data. These areas relate to oligotrophic oceanic waters, that are geographically large but grossly under-sampled. Overall, this figure provides quite robust evidence that the synthetic dataset has global coverage, from the clearest oceans until all kinds of coastal waters, and that the bio-optical relationships adopted in thus study are in line with empirical evidence.

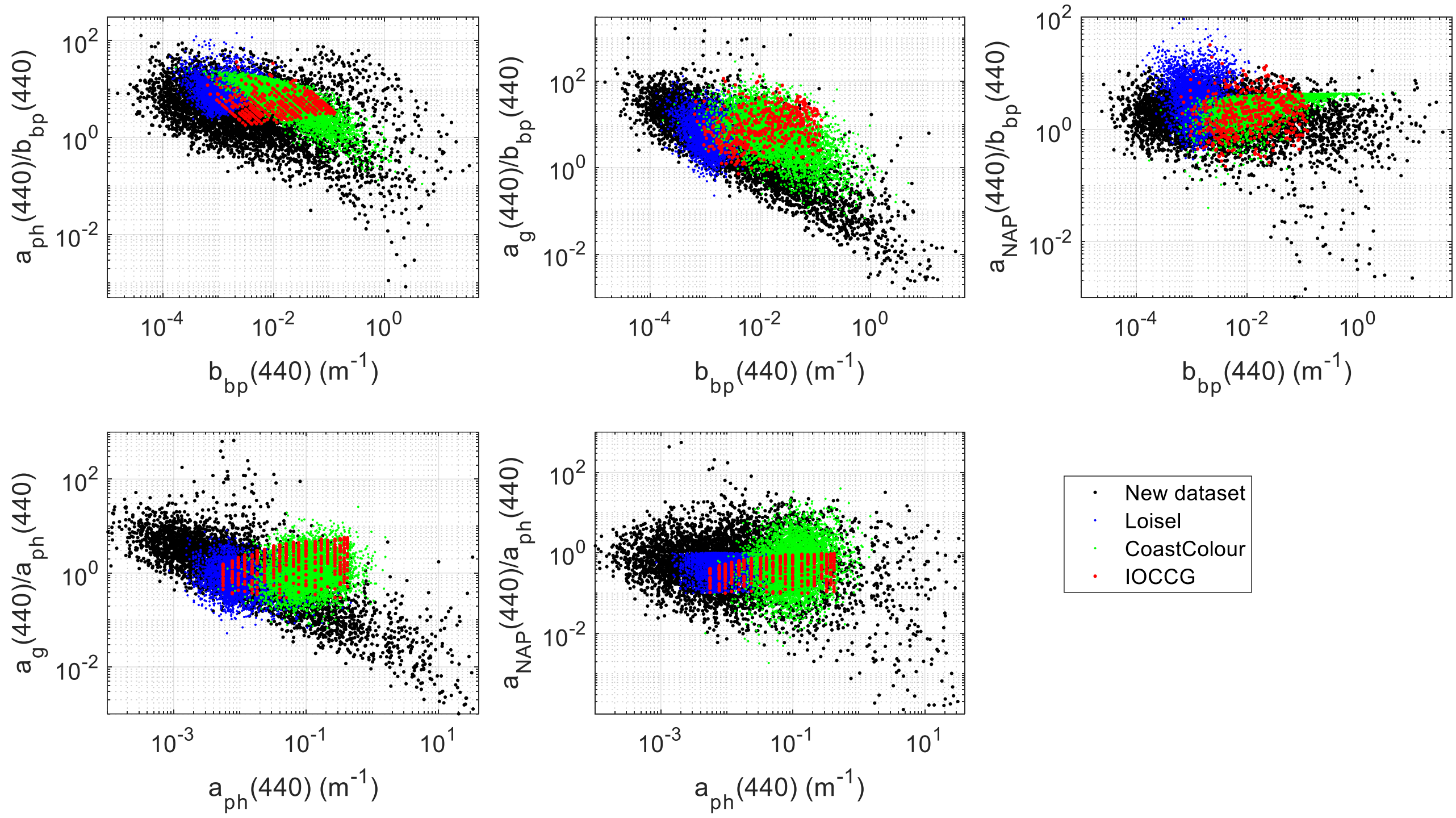

**Figure 11: IOP cross-relationship comparison between this and other synthetic datasets.**

This dataset is also compared to the three publicly available synthetic datasets in Fig. 11: the IOCCG dataset (IOCCG, 2006), the Coastcolour dataset (Nechad et al., 2015) and Loisel's dataset (Loisel et al., 2023). The plots highlight that the new dataset covers a much more diverse range of waters than other datasets, acknowledging that such datasets were not aimed at including the widest range of water as this dataset is. Some overlap in the publicly available datasets is noticeable for all crossed IOPs, with Loisel's dataset more shifted towards clearer waters than IOCCG and Coastcolour. Also, Loisel's dataset shows trends that appear more consistent to our dataset. As an example, $a_g(440)/a_{ph}(440)$ appear to show a general decreasing trend with $a_{ph}(440)$, corroborated with the in situ datasets. This is well reproduced with Loisel's dataset, whereas the IOCCG dataset shows the opposite trend. The Coastcolour dataset covers the upper part of the range, but due to its optical modelling, many dots are clustered near each other, instead of covering a wider range of values.

### 3.2 Radiative transfer calculations

Radiative transfer simulations were made with Hydrolight 5.1.2 (Sequoia Scientific, Inc.). The software was configured with a generic "case 2" water scenario, and the input IOP parameters were set as detailed in section 2.2. Inelastic scattering effects were not considered.

Normalized sky radiances were computed using the sky model "HCNRAD" (Harrison and Coombes Normalized RADiances) (Harrison and Coombes, 1988). Diffuse and Direct Sky irradiances were computed using the "RADTRANX" (RADTRAN eXtended for 300-1000 nm) model (Gregg and Carder, 1990). The ozone concentration was estimated from a climatology derived with binned monthly average TOMS v8 Ozone concentrations (data from 2000-2004 were averaged to give 5-year climatological averages for 5° latitude and 10° longitude quadrants), for the 90$^{th}$ day of the year, coordinates 40 ° N and 0 ° E, resulting in 354.9 Dobson units. The US Navy aerosol model was fed with the values: air mass type 5, relative humidity 80.0 %, precipitable water 2.5 cm and horizontal visibility 40.0 km. For the sea surface modelling, a wind speed of 5.0 m/s was assumed. Water index of refraction was calculated as a function of wavelength (Roettgers et al., 2016) for the given seawater T = 20.0 ºC and S = 35.0 PSU. The sea was considered homogeneous in depth and infinitely deep.

Phase functions are a critical component of bio-optical modelling if the angular variability of the light field is considered relevant. Here, phase functions from the Fournier-Forand (FF) family were used both for phytoplankton and for non-algal particles, as they fit very well the angular pattern of measured phase functions. Mobley et al. (2002) documented the indexing of the FF PFs as a function of the backscattering ratio only, a mechanism that is included in Hydrolight.

The source code of Hydrolight was modified so that the "printout" output files included the remote-sensing reflectance, both above and below the surface, for the whole set of viewing zenith and azimuth angles defined by Hydrolight default quadrants, that is, view angle varying from 0 to 80 ° in steps of 10 ° and then a last value of 87.5º (10 values in total), and azimuth varying from 0 to 180 ° in steps of 15 ° (13 values in total). Then, simulations were made for the whole range of sun zenith angles defined by the quadrants, that is from 0 to 80 ° in steps of 10 ° and then a last value of 87.5º (10 values in total). Therefore, for every IOP set up, directional AOPs are given at 1300 angles, and non-directional AOPs are given at the 10 sun zenith angles.

### 3.3 Reflectance overview and classification

Synthetic $R_{rs}$ were scrutinized to ensure that a diverse range of optical water types had been produced. The data underwent partitioning into twelve clusters via a k-means algorithm (Figure 12). Ternary plots were employed to visualize the absorption budget for all $R_{rs}$ within each class, with curves and dots colored based on particle backscattering. This classification is only used here as a method to show the extensive optical diversity within the dataset and does not constitute a part of the dataset. Descriptively, the following water types are:

- Classes 2 and 6 relate to clear oceanic waters.
- Class 1 corresponds to highly absorbing waters, with little NAP content.
- Classes 3,5, 7 and 8 represent coastal waters, exhibiting moderate concentrations of all constituents, in varying proportions.
- Classes 4 and 9 display highly productive waters, marked by high CDOM and NAP levels, respectively.
- Classes 10, 11 and 12 portray highly and very highly turbid waters. Notably, despite categorizing this water type into three classes, their cumulative occurrence is discrete. This outcome stems from the classification, which accentuates disparities in $R_{rs}$ values that are high.

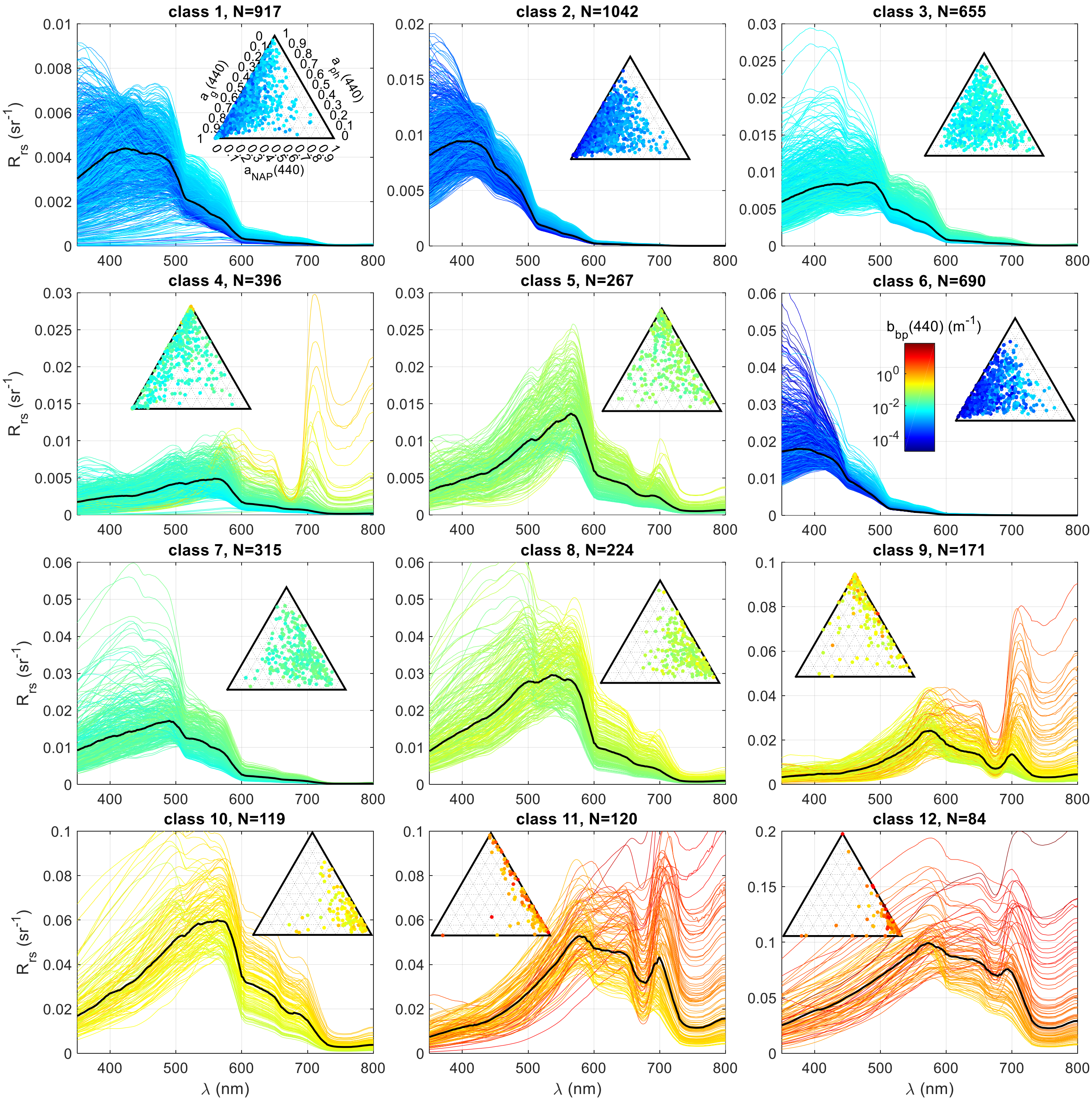

**Figure 12 *$R_{rs}$* spectra (normalized geometry) of the synthetic dataset, divided into twelve classes using the k-means classifying algorithm. Relative to each class, the ternary plots of the absorption budget are plotted. Line and dot color indicates particle backscattering at 440 nm, according to the attached color bar. Note varying vertical scale, across the classes, necessary to visualize the spectral variabilty across the dynamic ranges.**

### 3.4 Angular variation

Besides the wide IOP ranges, highlighted in the water classes of the following section, a unique characteristic of this dataset is the inclusion of the AOPs for the whole range of sun-view geometries. This matter is relevant for algorithm development and validation; for instance, in either in situ or satellite $R_{rs}$, the sun is very rarely at the zenith. The view angle is off nadir in above-water platforms and in satellite data, and the azimuth is normally such that avoids the maximum sun glint. This $R_{rs}$ bidirectionality is very often ignored. Algorithms that use band ratios, such as the oceanic OCx, partially suppress the bidirectional effect because the spectral behaviour is quite flat, but algorithms that rely on the absolute magnitude of $R_{rs}$ will inevitably propagate bidirectional effects as errors. This section showcases the anisotropy of $R_{rs}$ for two distinct water types. The first represents very oligotrophic oceanic waters, while the second could correspond to turbid areas with high CDOM, which can be found in shallow marginal seas such as the Azov Sea. The azimuthal angle definition follows that of Hydrolight (i.e., solar photons travel in the $\phi = 180\ °$ direction, that is, the sun is located at $\phi = 0$).

A first example of the $R_{rs}$ anisotropy for a clear water scenario is displayed in Fig. 13, for three wavelengths and five sun zenith angles. Related Fig. 14 focuses on one sun zenith angle, two solar azimuthal planes and a constant zenith view section. Increasing the sun zenith lowers the azimuthal symmetry and strengthens the radiance anisotropy. A zone of higher values forms along the solar plane for $\phi = 0$. It is known that for very clear waters, the single-scattering approximation can, at least qualitatively, explain the results. The phase functions of both water and particles have a local maximum at $\Psi = 180°$, leading to an overall maximum at $\theta = 60\ °$, that is, the back-scattering direction. The secondary maximum at $\theta = -60\ °$ (or $\theta = 60\ °$ for $\phi = 180\ °$) can be explained by the balance between progressive increase in the particle phase function and decrease in the water phase function as $\Psi$ decreases.

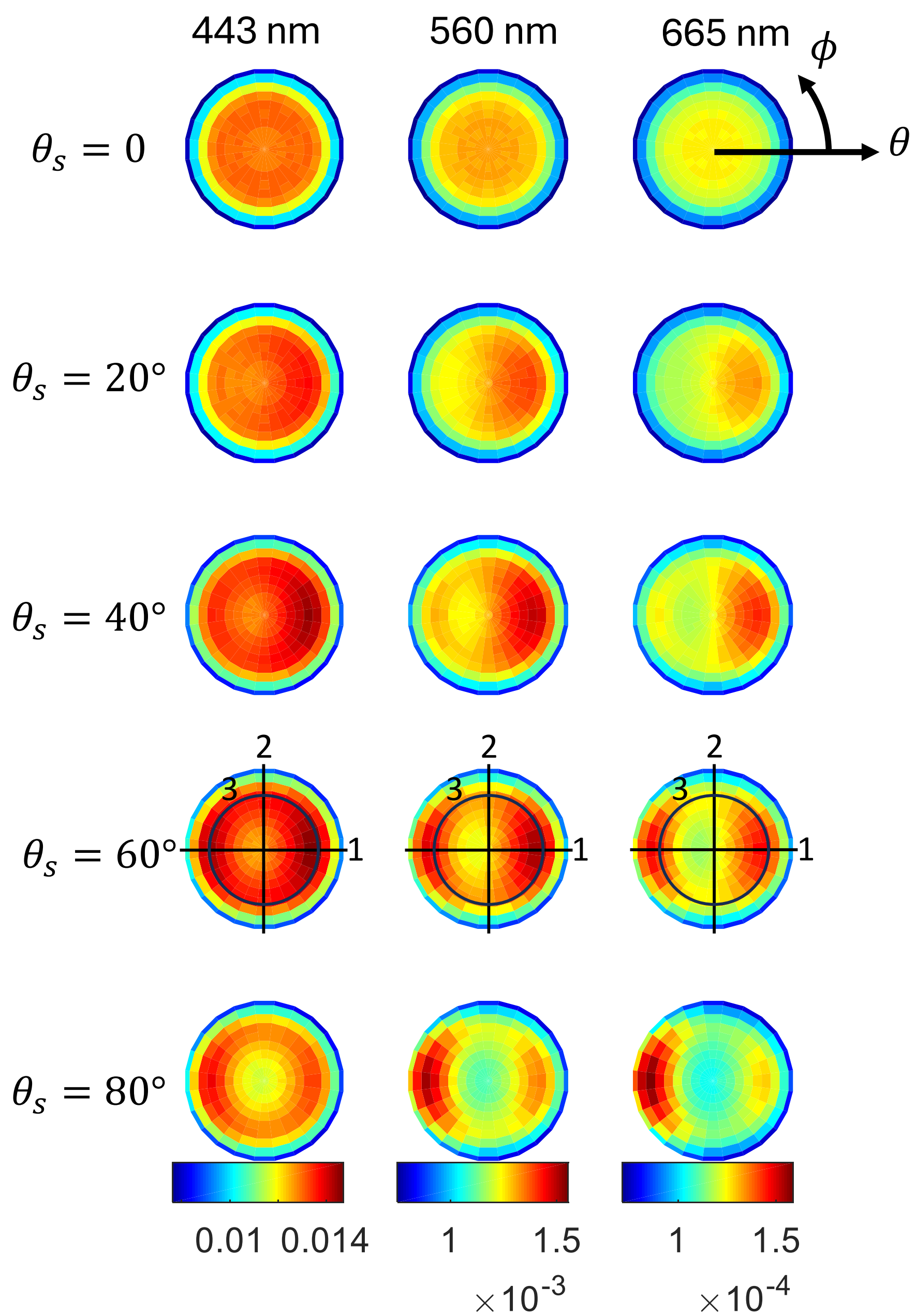

**Figure 13 Angular variability of $R_{rs}$ for the oligotrophic water spectrum shown in Figure 14. The polar plots are divided into selected sun zenith angles (rows) and wavelengths (columns). The polar angle represents the azimuth (zero "looking at the sun"), while the radius represents the radiance propagation angle (same as the viewing zenith angle). The color represents the $R_{rs}$ magnitude. The color scale among wavelengths for visualization purposes. For $\theta_s$=60° specifically, some indicated slices are presented in 1D plots in Figure 14. Section 1: sun's meridian plane. Section 2: perpendicular plane to the sun's meridian plane. Section 3: constant $\theta$=60°.**

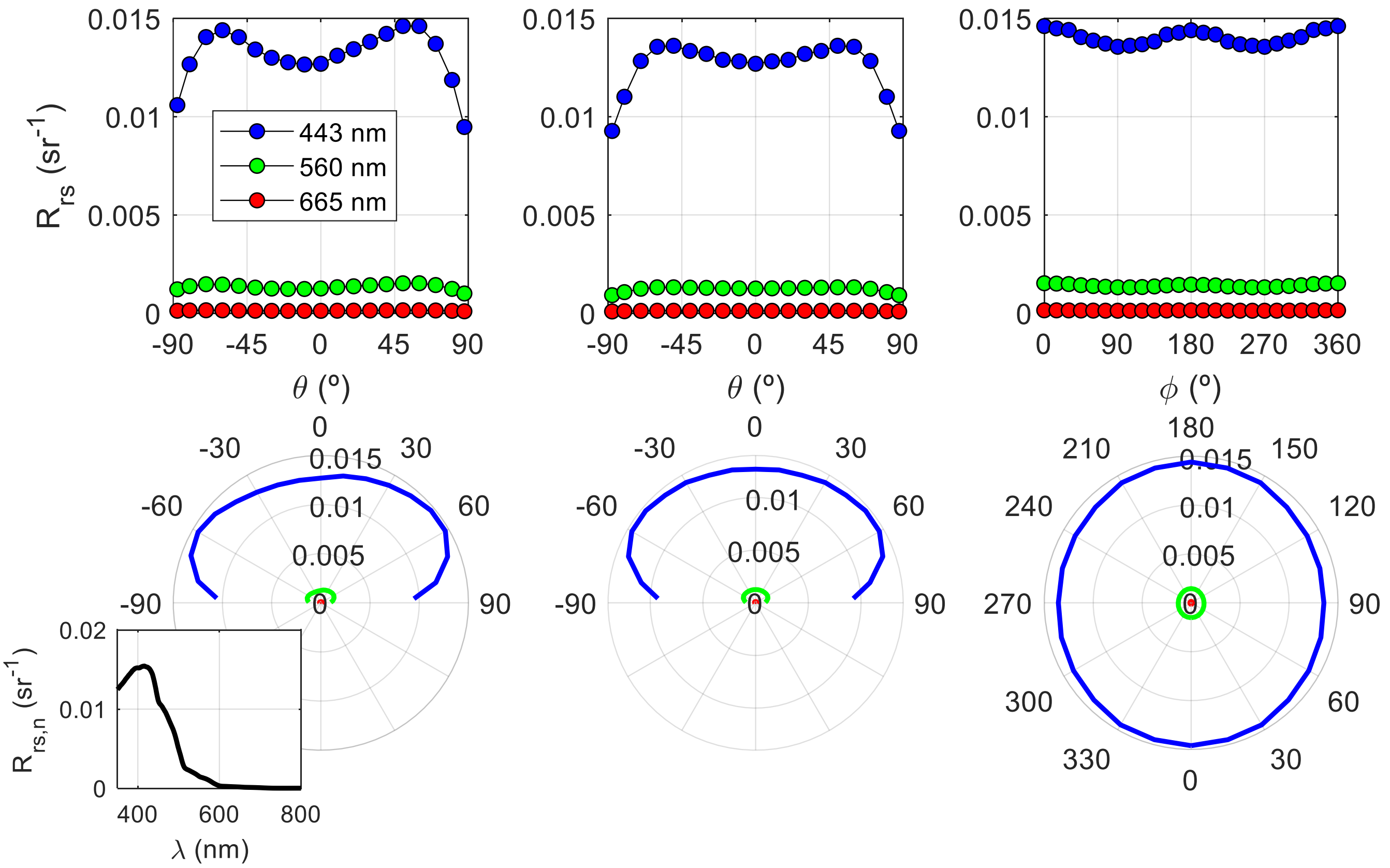

**Figure 14 Angular variability of Hydrolight-simulated $R_{rs}$ for the oligotrophic water case (spectrum shown in a corner). The plots represent the three sections for $\theta_s$=60° in Figure 13, in consecutive columns. Here, the sections are plotted in cartesian coordinates in the upper plots and polar coordinates in the lower ones.**

Figures 15 and 16 show an analog example for a turbid water scenario. Notable is the azimuthal maximum shifts to the $\phi = 180\,°$ direction. This is explained by the dominance of the particle phase function and the appearance of multiple scattering, which starts to become important even for small concentrations. This implies that the radiance at angle $\theta = -70\,°$ (or $\theta = 70\,°$ for $\phi = 180\,°$) is less influenced by the shape of the phase function at the particular direction given by the single scattering direction. Instead, multiple scattering does not randomize the light field in all directions, making it isotropic, but instead, makes the resulting radiances influenced by the phase function in variable ranges reaching $\Psi < 120\,°$, where it increases sharply. This behaviour was already documented (Loisel and Morel, 2001), but it was somehow not assimilated by most within the community.

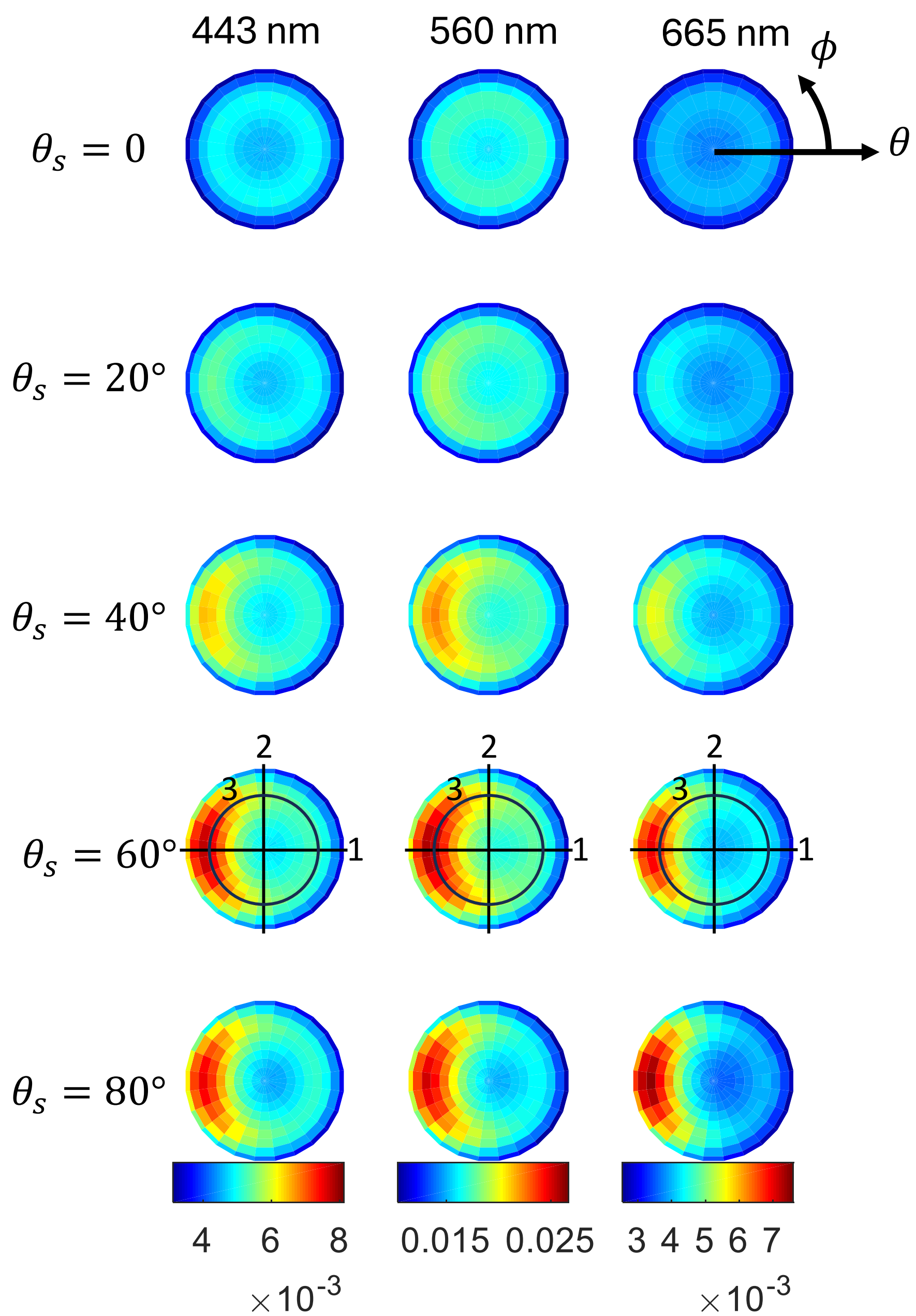

**Figure 15 As in Figure 13, but for the angular variability of $R_{rs}$ for the turbid waters.**

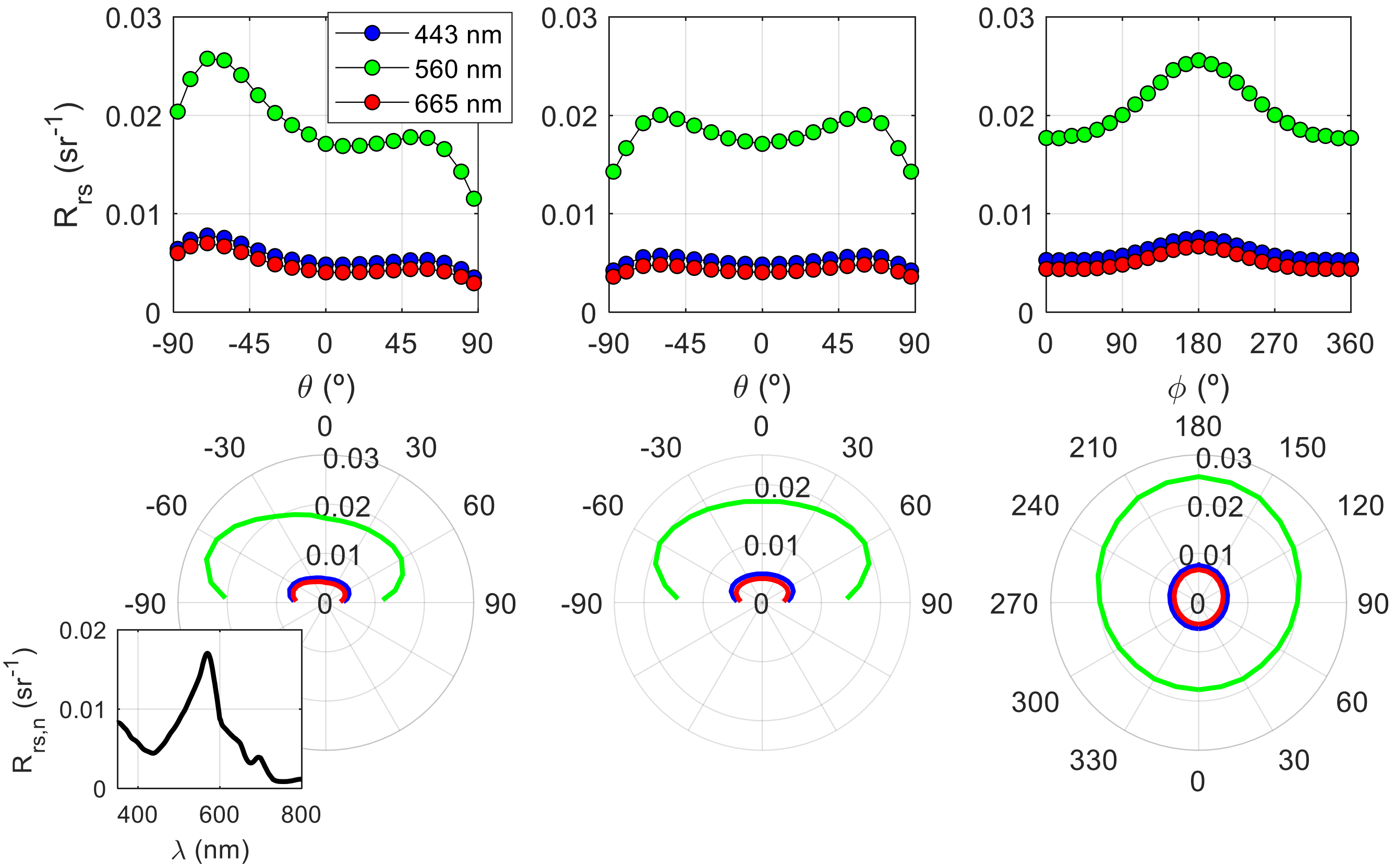

**Figure 16 As in Figure 13, but for the angular variability of $R_{rs}$ for the turbid waters.**

## 4. Data file description

Output data is organized in netCDF files, where each file contains a given IOP setup and all directional AOP output. Table 4 details the content of them. Variables have different sizes, according to their dependence on the following variables that can take the following number of different values: sun zenith angle $\theta_s, n_s = 10$, zenithal direction of radiance propagation, $\theta, n_\theta = 10$, azimuthal direction of radiance propagation $\phi, n_\phi = 13$, wavelength of radiation in vacuum $\lambda, n_\lambda = 451$. All in-water AOPs refer to the zero depth just below the surface. Diffuse attenuation coefficients instead required the choice of two depths approximate the depth derivatives, which were 0 m and 1 cm, as set by default in Hydrolight.

**Table 4 File description**

| Parameter | Description | Units | Size |
|---|---|---|---|

| | | | |
|---|---|---|---|
| C | Chlorophyll concentration | mg $m^{-3}$ | 1x1 |
| N | Non-algal particles concentration | g $m^{-3}$ | 1 x 1 |
| Y | Light absorption coefficient of coloured dissolved organic matter at 440 nm | $m^{-1}$ | 1 x 1 |
| theta_s | Sun zenith angle (zero at zenith) | º | $n_s$ x 1 |
| theta | Zenithal direction of radiance propagation (zero towards zenith) | º | $n_\theta$ x 1 |
| phi | Azimuthal direction of radiance propagation (zero towards the sun) | º | $n_\phi$ x 1 |
| lambda | Wavelength of radiation in vacuum | nm | $n_\lambda$ x 1 |
| Esdir_Es_ratio | Above-surface direct to total downwelling irradiance ratio | - | $n_s$ x $n_\lambda$ |
| aw | Spectral light absorption coefficient by seawater at 20 ºC and S=35 PSU | $m^{-1}$ | $n_\lambda$ x 1 |
| aph | Spectral light absorption coefficient by phytoplankton | $m^{-1}$ | $n_\lambda$ x 1 |
| ay | Spectral light absorption coefficient by coloured dissolved organic matter | $m^{-1}$ | $n_\lambda$ x 1 |
| aNAP | Spectral light absorption coefficient by non-algal particles | $m^{-1}$ | $n_\lambda$ x 1 |
| bw | Spectral light scattering coefficient by seawater at 20 ºC and S=35 PSU | $m^{-1}$ | $n_\lambda$ x 1 |
| bph | Spectral light scattering coefficient by phytoplankton | $m^{-1}$ | $n_\lambda$ x 1 |
| bNAP | Spectral light scattering coefficient by non-algal particles | $m^{-1}$ | $n_\lambda$ x 1 |

| | | | |
|---|---|---|---|
| bbw | Spectral light backscattering coefficient by seawater at 20 °C and S=35 PSU | $m^{-1}$ | $n_\lambda$ x 1 |
| bbph | Spectral light backscattering coefficient by phytoplankton | $m^{-1}$ | $n_\lambda$ x 1 |
| bbNAP | Spectral light backscattering coefficient by non-algal particles | $m^{-1}$ | $n_\lambda$ x 1 |
| Rrs | Spectral angle-dependent above-water remote sensing reflectance $\left(\frac{L_w}{E_s}\right)$ | $sr^{-1}$ | $n_s$ x $n_\theta$ x $n_\phi$ x $n_\lambda$ |
| rrs | Spectral angle-dependent underwater radiance reflectance $\left(\frac{L_u}{E_d}\right)$ | $sr^{-1}$ | $n_s$ x $n_\theta$ x $n_\phi$ x $n_\lambda$ |
| Q | Spectral angle-dependent underwater Q-factor $(\frac{E_u}{L_u})$ | sr | $n_s$ x $n_\theta$ x $n_\phi$ x $n_\lambda$ |
| Kou | Spectral diffuse attenuation coefficient of scalar upwelling irradiance | $m^{-1}$ | $n_s$ x $n_\lambda$ |
| Kod | Spectral diffuse attenuation coefficient of scalar downwelling irradiance | $m^{-1}$ | $n_s$ x $n_\lambda$ |
| Ko | Spectral diffuse attenuation coefficient of scalar total (spherical) irradiance | $m^{-1}$ | $n_s$ x $n_\lambda$ |
| Ku | Spectral diffuse attenuation coefficient of planar upwelling irradiance | $m^{-1}$ | $n_s$ x $n_\lambda$ |
| Kd | Spectral diffuse attenuation coefficient of planar downwelling irradiance | $m^{-1}$ | $n_s$ x $n_\lambda$ |
| Knet | Spectral diffuse attenuation coefficient of net planar irradiance | $m^{-1}$ | $n_s$ x $n_\lambda$ |
| KLu | Spectral diffuse attenuation coefficient of upwelling radiance towards the zenith | $m^{-1}$ | $n_s$ x $n_\lambda$ |

| | | | |
|---|---|---|---|
| mu_u | Spectral average cosine of the upwelling radiance | - | $n_s$ x $n_\lambda$ |
| mu_d | Spectral average cosine of the downwelling radiance | - | $n_s$ x $n_\lambda$ |
| mu_tot | Spectral average cosine of the total radiance | - | $n_s$ x $n_\lambda$ |
| R | Spectral underwater irradiance reflectance $(\frac{E_u}{E_d})$ | - | $n_s$ x $n_\lambda$ |

## 5. Data availability

Data described in this manuscript can be accessed at Zenodo under https://zenodo.org/records/11637178 (Pitarch and Brando, 2024). The repository hosts two versions of the dataset: one hyperspectral, from 350 nm to 900 nm, in steps of 1 nm, and a smaller, multispectral on, for the twelve Sentinel 3-OLCI bands between 400 nm and 753 nm.

## 6. Conclusions

With the development of the presented synthetic dataset, encompassing inherent and apparent optical properties alongside associated optically active constituents, we believe to have filled several gaps as identified in our literature review of publicly available in situ and synthetic datasets. On one hand, the large quantity and high quality of the in situ data allowed the application of stringent quality control procedures to develop novel bio-optical relationships involving parameters that model absorption and scattering of the optically active constituents. The spread in the data clouds used for bio-optical modelling was reproduced as probability density functions, resulting in a realistic depiction in the synthetic dataset of the natural variability of the in situ data. Our dataset is therefore representative of waters of varying trophic levels and optical complexity. As a by-product of the underlying the reported bio-optical relationships can be assumed to become a reference for future optical studies.

Apparent optical properties are resolved at all geometric angles available by the radiative transfer simulations, making this one the first directional dataset ever published. This detail makes it suitable for directional studies of reflectance, diffuse attenuation and any other derived quantity.

The synthetic dataset is distributed in netCDF files as it is convenient for data storage and space management. Given the very fine spectral resolution of 1 nm between 350 nm and 800 nm and that each file contains the IOP setup as well as all directional AOPs for all 1300 angular configurations (and hemispheric variables such as $K_d$ are included for all 10 sun zenith angles), each of the 5000 files only weights approximately 5700 kB. The netCDF format also makes the dataset easy to handle using common software packages.

## 7. Author contribution

J.P., V.E.B: Conceptualization of the study, development or design of methodology, validation, Writing – review & editing. J.P.: Data curation, Formal analysis, Software, Visualization, Writing – original draft preparation. V.E.B.: Funding acquisition, Project administration.

## 8. Competing interests

The authors declare that they have no conflict of interest.

## 9. Acknowledgements

We are grateful to Davide d’Alimonte, Tamito Kajiyama, Constant Mazeran and Marco Talone for carrying out independent analyses with previous versions of this dataset, that were fundamental to develop it to its final configuration. Flavio la Padula and Vega Forneris assisted with IT requirements.

This study was carried out in the frame of the Copernicus study “BRDF correction of S3 OLCI water reflectance products” (contract No.RB_EUM-CO-21-4600002626-JIG), conducted by EUMETSAT. The work also acknowledges the support of the Ocean Colour Thematic Assembly Centre of the Copernicus Marine Environment and Monitoring Service (contract: 21001L02-COP-TAC OC-2200–Lot 2: Provision of Ocean Colour Observation Products (OC-TAC)). J.P. thanks financial support by the EU - Next

Generation EU Mission 4 "Education and Research" - Project IR0000032 – ITINERIS - Italian Integrated Environmental Research Infrastructures System - CUP B53C22002150006.

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
