# Peer review of "A hyperspectral and multi-angular synthetic dataset for algorithm development in waters of varying trophic levels and optical complexity"

_Earth System Science Data, 2024_

## Referee Comment (RC1)

**Review of essd-2024-295: "A hyperspectral and multi-angular synthetic dataset for algorithm development in waters of varying trophic levels and optical complexity" by Jaime Pitarch, Vittorio Ernesto Brando**

General comment:

This manuscript details the generation of synthetic dataset of the "apparent optical properties" (AOP) of different water types based on radiative transfer (RT) computations. The authors collated a large set of "inherent optical properties" (IOP) from the literature on which they applied statistical treatment to encompass the natural diversity (and cross-correlation) of the absorption and scattering features of the optically active component of the water column. This effort is a prerequisite to further study the (non-linear) relationships between IOP and AOP in order to advance in remote sensing algorithm development. Nevertheless, such a dataset should have been validated on actual data (e.g. optical closure) to evaluate the numerous assumption/approximation of the spectral IOP (e.g., phase function, spectral slope of the scattering coefficient, specific absorption/scattering…). In the present form, the computed AOP cannot be considered as a reference dataset to be shared as is to the scientific community. The manuscript is closer to a research paper than a data paper. The computations are of interest and should deserve deeper analysis especially on the anisotropy properties of the remote sensing reflectance in a dedicated research paper before publication as a "data paper".

Specific comments:

The manuscript structure could be significantly improved to introduce the terms/equations before their usage in the text body (hard to follow in the present shape). I would advise to start with the radiative transfer equation and detail the terms to be used as input to solve the equation to provide the AOP outcomes

L.11: "optical domain" is very large, maybe, replace with "UV-visible to near-infrared range".

L.18: from 350 nm to ??? nm (give the upper limit)

L.40 and L.42: please avoid to use "C" and "N" which could be understand by Carbon and Nitrate in the biogeochemical community

L.130-131: $B_{ph}$ not defined. Why using "ph" for both whereas the latter is for "non-algal"

L.152: "A" and "E" not defined

L.154: "spectral slope" not defined

L.155: "..scattering coef. are set constant": you mean spectrally constant?

L. 181-183: $B_{ph}$, $b_{ph}$, $b_{b,ph}$ : not defined

L.188: "more realistic angular variation", is this statement demonstrated, any reference?

L. 229: you could introduce here the concept of "specific absorption", it would be nice as well to discuss the impacts of different specific absorption coefficients in your study (e.g., different phytoplankton species/mixture)

L.256 and 260: could you elaborate on the choice of the thresholding values

L.272: it would be of interest to show the phytoplankton specific absorption in a dedicated figure.

L. 310: "the offset was removed"... be careful with this assumption that is true for CDOM but not for minerals that might absorb in the red and near-infrared

L.312: title "particle backscattering" is for phytoplankton and non-algal particles/ minerals?

Table 1: several terms are not defined for instance U, N , etc. that I understand as being uniform and normal distributions… but you used N for concentration of non-algal particles

L. 435: why using h=0.7 ??

L. 450 and 457: "a_ph < c_ph which is unphysical", I think you would mean the opposite "a_phy > c_phy …

L. 459-464: on the setting of "B_ph", I think in your study this setting is critical to compute the angular shape of the VSF and therefore the impact of the anisotropy of the remote sensing reflectance (e.g. BRDF)

Fig 5: x-label "C/N" very close to carbon to nitrate ratio….

L. 704: "1300 angles", could you give the range (and increment) of the viewing zenith and azimuth angles

L.748: letter "Psi" not definedL. 765: "... for a turbid water scenario", could you give the sediment (SPM) concentration

"

---

## Author Comment (AC3)

The manuscript by Pitarch and Brando proposes a comprehensive data set of IOP and AOP values from radiative transfer simulations and related parameterisations relying on previous scientific investigations and high quality in situ data.The newly proposed data set definitively shows advances with respect to various predecessors and because of this it deserves to be supported by the proposed manuscript.

We really appreciate Giuseppe Zibordi's comments on the paper. We believe to have generated a dataset that redefines the state of the art of bio-optical modelling. Also, this dataset is provided across a much-extended range of water types respect to past attempts. In addition, the AOPs are angularly resolved, a feature that makes the dataset unique respect to its predecessors, and makes it attractive for bidirectional studies, that are not only needed for current sensors, but that will be facilitated by new satellites that incorporate multiangular sensors, such as PACE.

I only have three comments I would like to convey to the authors.

The first is quite minor and refers to the terminology. The term ' synthetic ' should be replaced by the more appropriate ' simulated '.

We understand this comment. There is no total agreement in the community on which term to use and both have been used. We have seen both terms being used. Here, we follow the terminology started by the publication of the related IOCCG report, which mentioned a *synthetic* dataset.

The second on the statement qualifying simulated data not affected by errors. This is quite questionable: simulated data can only provide an 'interpretation' of the 'truth' based on a number of input parameters and modelling solutions. Regardless of the RT solution, the input parameters may not capture the actual 'truth'.

This comment has the point, and we are willing to upload a revised version of the manuscript where there are enhanced verifications on the representativeness of the generated RT simulations, based on the comparison between $R_{rs}$ and absorption, chlorophyll and TSM. It will be shown that our dataset represents all water types, in terms of the covariability of the related variables for the in situ datasets available.

The last comment is the most relevant one. It is commendable that the data set is proposed with 1-nm spectral resolution. However, it is questionable that the simulated data can actually capture 1-nm spectral variations. This appears confirmed by the aggressive smoothing applied to the experimental aph values. This limitation should be acknowledged.

Raw $a_{ph}$ measurements are noisy. Sometimes, the smoothing is made by whoever provides the data but, in most cases, we found evident noise. Such noise would influence the simulated AOPs if not filtered out, hence the aggressive smoothing. This does not imply *per se* that the 1 nm spectral resolution is lost after this process, as it is well accepted that an $a_{ph}$ results from the combination of pigments that have smooth features. However, we are willing to provide some lines of discussion on the difficulties of the in situ determination of an $a_{ph}$ spectrum, and on the potential limitations for the spectral resolution.

Giuseppe Zibordi

---

## Author Response (AR1)

Dear Editor,

We hereby resubmit our manuscript *A hyperspectral and multi-angular synthetic dataset for algorithm development in waters of varying trophic levels and optical complexity* for publication in ESSD.

You provided two reviews of the manuscript, whose comments we have addressed in the revised version. From their comments, there were a number of major requested changes and we have addressed all of them. Specifically:

- *Validation with in situ radiometric data*: we have added a new section where the remote sensing reflectance of our dataset is compared to some water properties, like the absorption coefficient, the chlorophyll concentration and the total suspended matter concentration. This is done for our synthetic dataset compared to independent in situ dataset. The good agreement of the trends for all cases provides the confidence in the adequate representativity of our dataset for natural waters. For this sake, we have incorporated the in situ dataset by Zibordi and Berthon (2024), that has appeared very recently, to help our validation exercises. In addition, we have calculated a quality index for our dataset, that was developed from in situ data. Again, the results are satisfactory.
- *Need to introduce variables and symbols*: we have moved a section that introduces many optical variables to the first section. Besides that, we have ensured that all variables are now properly introduced before use.
- *Some additional context in the presentation of modelling choices*: it is acknowledged now that there are some steps in our optical modelling where other options could be considered, and in the revised version we put our work more into context.
- *Clarification on data range*: from the previous version, readers might interpret that there was significant range extension beyond natural limits in the lower part of the range, specifically in what regards CDOM. With new plots and clarification, we now make clear that there is no such extension, and that our dataset covers data ranges that are documented in situ, as well as in other datasets, in this part of the range.

Detailed comments to both reviewers were already published as responses but are also attached here.

We want to underline that there were three community comments to our manuscript from leading experts in our field, all of them recognizing the importance of our dataset for an array of optical studies and supporting the publication of the manuscript.

In addition to all changes, we have also taken the opportunity to rephrase many sentences in the manuscript to improve the quality and readability. There is also a simplification of the section structure in what regards data and methods.

All changes are properly tracked.

We believe that you will appreciate the quality of the submitted version.

Best regards,

Jaime Pitarch (corresponding author).

**Review of essd-2024-295: "A hyperspectral and multi-angular synthetic dataset for algorithm development in waters of varying trophic levels and optical complexity" by Jaime Pitarch, Vittorio Ernesto Brando**

**General comment:**

This manuscript details the generation of synthetic dataset of the "apparent optical properties" (AOP) of different water types based on radiative transfer (RT) computations. The authors collated a large set of "inherent optical properties" (IOP) from the literature on which they applied statistical treatment to encompass the natural diversity (and cross-correlation) of the absorption and scattering features of the optically active component of the water column. This effort is a prerequisite to further study the (non-linear) relationships between IOP and AOP in order to advance in remote sensing algorithm development.

We share the reviewer's summary and we believe that our manuscript contributes to the advance. It is correct that a large amount of effort was put into the collection of a very large in situ dataset for the elaboration of needed bio-optical relationships and the verification of crossed relationships among many parameters. The statistical treatment that is proposed to the inherent optical properties and the optically active constituents is a significant step forward too.

Nevertheless, such a dataset should have been validated on actual data (e.g. optical closure) to evaluate the numerous assumption/approximation of the spectral IOP (e.g., phase function, spectral slope of the scattering coefficient, specific absorption/scattering...). In the present form, the computed AOP cannot be considered as a reference dataset to be shared as is to the scientific community.

Part of this comment appears to ignore the background on the development of synthetic datasets, some of them published in ESSD (Loisel et al. 2023; Nechad et al. 2015), as well as previous research on phase functions and phase modelling. The paper already uses a multitude of actual bio-optical data to evaluate numerous assumptions and approximations. This includes the verification of the phase function for phytoplankton (Fig. 4), the spectral slope of the backscattering coefficient for NAP (Fig. 8) and specific absorption or scattering (Figs. 2,3,4,5,7). Other assumptions are well supported by the most recent literature. The reviewer will note that, among the generated synthetic dataset, this is the one with the most careful evaluation of all the involved relationships, where every single step has been given proper justification, compared to previous ones (Loisel et al. 2023; Nechad et al. 2015).

Optical closure is the comparison between measured AOPs and synthetic ones, from a concomitant set of IOPs. This procedure is valid to ensure that a given bio-optical dataset measured in the field is considered self-consistent. It also can be used to evaluate several data reduction methods for IOPs (Pitarch et al. 2016; Tonizzo et al. 2017). In a more generic way, it can be interpreted as the comparison between several methods to provide analogue quantities. We interpret the request for optical closure as the need to compare the parameterization as well as the synthetic dataset with actual data. After reading this review and the reviewer's reply to other comments, we guessed the manuscript was missing some crossed plots between $R_{rs}$ and some measured water constituents, both for our synthetic dataset and for independent empirical data. We have verified that such plots were present in previous reference dataset description paper in optics (Loisel et al. 2023; Nechad et al. 2015). Hence, to keep the same quality standard here, we decided to inspect such relationships and incorporate them in the revised version. In addition, we have also evaluated our generated reflectances through the spectral quality index recently developed by Dierssen et al. (2022), to provide further confidence on the reliability of the dataset. Given that these are four plots in total, they will probably need their own subsection. They are also included in this rebuttal below.

[Figure]

Figure R1.1 Upper plot: scatter plot between the apparent optical wavelength (Vandermeulen et al. 2020) and the NDI index: $NDI(492,665) = \frac{R_{rs}(665) - R_{rs}(492)}{R_{rs}(665) + R_{rs}(492)}$. Magenta lines: QWIP score (Dierssen et al. 2022) and error bars. Lower plot: histogram of the QWIP score, defined as the difference respect to the QWIP curve.

To generate Figure R1.1, we calculated the QWIP index by Dierssen et al. (2022) for our entire synthetic dataset. Such index aims at providing a quality estimate for a hyperspectral $R_{rs}$. QWIP was developed a large dataset of in situ $R_{rs}$, so this comparison is actually a comparison with real $R_{rs}$ data. In Dierssen et al. (2022), it is mentioned that values within the 0.2 margins have high similarity to real spectra measured in the field, which are all 5000 but 7 spectra. Still, these 7 spectra are close to the limit, and may simply contain some bio-optical characteristics, not present in the QWIP calibration dataset. This comparison, therefore, gives confidence in the quality of our dataset.

[Figure]

Figure R1.2 A scatter plot between the $R_{rs}$-generated $\chi$ index and the matched non-water absorption spectrum at 560 nm $a_{nw}(560)$. Black dots are from the synthetic dataset and coloured dots are from field data from various references (see text).

Figure R1.2 helps to assess the covariability of $R_{rs}$ and the absorption coefficient. A one-dimensional predictor $\chi$ is derived from an $R_{rs}$:

$$\chi = \log_{10}\left( \frac{R_{rs}(443) + R_{rs}(490)}{R_{rs}(560) + 5\frac{R_{rs}^2(665)}{R_{rs}(490)}} \right)$$

This $\chi$ index is matched to non-water absorption spectrum at 560 nm $a_{nw}(560)$. There are several open access, freely available in situ datasets that contain both measured variables matched together, such as Valente et al. (2022), Zibordi and Berthon (2024) and the Schaeffer, Mouw and Biosope datasets (Casey et al. 2020). Figure R1.2 clearly shows the excellent average overlap between our synthetic dataset and measured data. Different bio-optical characteristics produce slight deviations from the mean curve, indicating natural variability.

[Figure]

Figure R1.3 Chlorophyll concentration as a function of the maximum band ratio for OC4-type algorithms, for the synthetic dataset and for data in Valente et al. (2022) and Zibordi and Berthon (2024).

Figure R1.3 shows how a given chlorophyll concentration in the dataset relates to the generated $R_{rs}$ through an index that is used to estimate chlorophyll in the ocean:

$$MBR_{OC4} = \frac{\max\left[R_{rs}(443), R_{rs}(490), R_{rs}(510)\right]}{R_{rs}(560)}$$

From $R_{rs}$, we calculate the maximum band ratio $MBR_{OC4}$, an index known to be a good predictor for its good correlation to chlorophyll concentration (C) in oceanic waters, but also used for studying the consistency of a given dataset in all kinds of water (Nechad et al. 2015). Here, matched $MBR_{OC4}$ and chlorophyll concentration from two large in situ datasets are plotted (Valente et al. 2022; Zibordi and Berthon 2024), showing a good general overlap, though with some degree of differences among them, that are explainable due to different bio-optical characteristics of the seas sampled. Data from our dataset

generally agrees with the trend.

[Figure]

Figure R1.4 Total suspended matter concentration as a function of $R_{rs}(665)$, for the synthetic dataset and for data in Valente et al. (2022) and Zibordi and Berthon (2024).

The last comparison to real $R_{rs}$ data involves the relationship to the total suspended matter concentration (T), relevant for coastal and inland water, which usually show higher turbidities. Our dataset does not need T for its generation, but it can be estimated as T=N+0.07C, after Brando and Dekker (2003), where N is the concentration of non-algal particles. It is known that T covaries with $R_{rs}$ at long wavelengths, and 665 nm is commonly employed, due to the lesser disturbance by CDOM. Figure R1.4 shows that our dataset has a range of natural variability that includes that in in situ datasets (Valente et al. 2022; Zibordi and Berthon 2024), once more confirming the suitability of this new dataset for optical studies in all ranges of water.

The reader must note that for the new plots discussed above, the dot cloud amplitude in the in situ datasets is included in the synthetic dataset, meaning that the statistical treatment that was given to the inherent optical properties prior to radiative transfer simulations was such to ensure optical representativeness of many water types, as far as this plot is concerned.

The manuscript is closer to a research paper than a data paper.

This paper has the classical structure of a data paper, as outlined by the journal policies. These mention that a data description paper must contain "original data", which our paper contains, and "an accurate account of the research performed", which also contains. In other words, although this one is a data paper, it still has to document some amount of research that was needed in order to generate the data. It follows the same structure of similar papers published recently on the same topic on ESSD (Loisel et al. 2023; Nechad et al. 2015).

The computations are of interest and should deserve deeper analysis especially on the anisotropy properties of the remote sensing reflectance in a dedicated research paper before publication as a "data paper".

We have generated a dataset with the additional novelty of directional variation. Any research derived from the usage of this dataset should come afterwards and not before the publication of this dataset. In terms of data quality, the dedicated analysis of the anisotropic properties of the remote sensing reflectance are a consequence of the bio-optical modelling, and strongly influenced by the phase function. Research has shown that the best choice for the phase functions to be used in radiative transfer simulation in natural waters are of the Fournier-Forand family (Berthon et al. 2007; Mobley et al. 2002; Pitarch et al. 2016; Sullivan and Twardowski 2009), and we make this choice in the paper, also in line with recent datasets in ESSD (Loisel et al. 2023). Once the bio-optical modelling is considered accurate enough, the angular variability of the reflectance generated by Hydrolight can be considered accurate. Here, the manuscript already includes figures displaying the angular variability, explained based on theoretical considerations and former results by Park and Ruddick (2005).

Research on the anisotropy of the reflectance supported by this dataset was carried out in the framework of an EUMETSAT project on BRDF correction for OLCI, and we already have two manuscripts in submission and in preparation. We can simply anticipate that if a semianalytical model is calibrated using this dataset, the directional modelling of the reflectance performs better than any existing model when compared to multiangular radiometric data.

**Specific comments:**

The manuscript structure could be significantly improved to introduce the terms/equations before their usage in the text body (hard to follow in the present shape).

We will make sure that every quantity is properly introduced.

I would advise to start with the radiative transfer equation and detail the terms to be used as input to solve the equation to provide the AOP outcomes

The manuscript was structured similarly to other papers that include Hydrolight simulations (e.g., Brando et al. 2012; IOCCG 2006; Loisel et al. 2023; Nechad et al. 2015; Pitarch et al. 2016), where there is no need to write the radiative transfer equation. But we will make sure that all concepts are clear.

L.11: "optical domain" is very large, maybe, replace with "UV-visible to near-infrared range".

We acknowledge that "optical domain" is not completely precise. We will use "extended optical domain", clearly making explicit the range.

L.18: from 350 nm to ??? nm (give the upper limit)

This is not the range of output data but the range of input phytoplankton spectra. The upper limit of a phytoplankton absorption spectrum is not relevant as far as it covers up to a minimum of ~720 nm, beyond which it is essentially zero and does not contribute to the absorption budget. To avoid lengthy descriptions, we prefer to keep it as it is.

L.40 and L.42: please avoid to use "C" and "N" which could be understand by Carbon and Nitrate in the biogeochemical community

In the field of optics, there is not yet a community consensus on which letters to use for these quantities. The reason to use such letters was to find a notation as compact as possible, given that they are used in many equations and plots. The author is right that C and N stand for the elements Carbon and Nitrogen in

chemistry, but it should be acknowledged that oftentimes there are variables or quantities that use similar letters across research fields. We therefore keep our notation.

L.130-131: B_ph not defined. Why using "ph" for both whereas the latter is for "non-algal"

That was a typo. The second $B_{ph}$ will be replaced by $B_{NAP}$.

Regarding the use of "ph" for "non-algal", the reviewers is right. This was a typo and will be corrected.

L.152: "A" and "E" not defined

We will provide an in-line definition by explicitly writing their defining equation $a_{ph}(\lambda) = A(\lambda)C^{E(\lambda)}$ after Bricaud et al. (1995).

L.154: "spectral slope" not defined

The concept "spectral slope" is used in bio-optical literature to mention an exponent or a coefficient within an exponent. We will cite Table 1, where they are properly put in context.

L.155: "..scattering coef. are set constant": you mean spectrally constant?

It is meant that they remain unchanged for the whole dataset (not this dataset but others). If they are spectra, they remain the same. This is a fact that we criticize because leaving these parameters constant hinders optical variability and does not agree with observations, as we report in several figures in the paper. We will try to clarify this sentence in the revised version.

L. 181-183: B_ph, b_ph,b_b,ph : not defined

We will make sure that these and other quantities are defined before usage in the paper.

L.188: "more realistic angular variation", is this statement demonstrated, any reference?

Yes. Numerous studies have demonstrated that Fournier-Forand phase function are best analytical representation at hand to model the angular scattering of marine particles, either from indirect evidence following simulations (Mobley et al. 2002; Pitarch et al. 2016) or from direct comparison to measured phase functions (Berthon et al. 2007; Sullivan and Twardowski 2009). We will incorporate relevant citations into this part of the manuscript.

L. 229: you could introduce here the concept of "specific absorption", it would be nice as well to discuss the impacts of different specific absorption coefficients in your study (e.g., different phytoplankton species/mixture)

The concept of "specific absorption" is not used in our bio-optical modelling. The reason is that specific absorption is defined as the absorption divided by a given concentration, in this case, chlorophyll, so: $a^*_{ph} = \frac{a_{ph}}{C}$. The problem with this concept is that it assumes that the $a_{ph}$ is proportional to $C$, which it is not realistic, as there is the phenomenon called packaging effect (Bricaud et al. 1995), which essentially implies that the increase of $a_{ph}$ with $C$ is less than proportional. For such a reason, we modelled $a_{ph}$ using a more complex approach that is a refinement of the procedure used to generate $a_{ph}$ for the IOCCG dataset (IOCCG 2006). We believe that the manuscript explains the procedure well enough. This approach is based on scaled $a_{ph}$ spectra that are picked from a large compilation of $a_{ph}$. Therefore, many phytoplankton mixtures are realistically represented.

L.256 and 260: could you elaborate on the choice of the thresholding values

Such thresholds were selected by experience, and trial-and-error. Essentially, in situ measurements of $a_{ph}$ spectra can sometimes be noisy or problematic depending on the equipment and the type of water

sampled, and these problems increase in the UV region. The dataset we gathered of $a_{ph}$ included some spectra that had suspicious features, i.e., high increase in the UV, many negative values, or spectral misalignment. We believe that our thresholds effectively kept a large number of high-quality spectra. These explanations are provided in the paragraphs from line 263 to 271.

L.272: it would be of interest to show the phytoplankton specific absorption in a dedicated figure.

See the related comment above.

L. 310: "the offset was removed"... be careful with this assumption that is true for CDOM but not for minerals that might absorb in the red and near-infrared

The reviewer has the point that. Although the NIR offset removal is well established for CDOM, absorption of mineral particles at the NIR may be real. While this is plausible, the community has not yet found ways to predict such an offset as a function of other parameters within the bio-optical modelling. For this reason, we decided to adopt a more conservative approach and model NAP absorption as an exponential that tends to zero in the NIR, following other dataset descriptions (IOCCG 2006; Loisel et al. 2023; Nechad et al. 2015). Extended explanations will be added in the text.

L.312: title "particle backscattering" is for phytoplankton and non-algal particles/ minerals?

Yes, particle backscattering includes backscattering by all particles other than water, and therefore, $b_{bp} = b_{b,ph} + b_{b,NAP}$. That is specified later in eq. (6).

Table 1: several terms are not defined for instance U, N , etc. that I understand as being uniform and normal distributions… but you used N for concentration of non-algal particles

Yes, in the statistical modelling in this manuscript U and N are the uniform and normal distributions, respectively. For N, we acknowledge some potential confusion to the N representing the concentration of non-algal particles, so this will be clarified wherever it is needed. We might consider using another font for the probability density functions.

L. 435: why using h=0.7 ??

This part of the bio-optical modelling is the one where a high number of assumptions must be made because there are no measurements of phytoplankton scattering or attenuation as a function of chlorophyll concentration. Therefore, for the principle of parsimony, we remained close to the modelling in the IOCCG dataset. However, we noticed that using $h = 0.63$ generated many realizations where $c_{ph} < a_{ph}$ (note that the modelling has a stochastic component). Noting that the CoastColour dataset (Nechad et al. 2015) has $h = 0.795$, we decided to raise $h$ just a bit until 0.7, which falls in between both, and finally allowed us to obtain realistic IOPs. Explanations for this choice will be given.

L. 450 and 457: "a_ph < c_ph which is unphysical", I think you would mean the opposite "a_phy > c_phy …

We thank the reviewer for finding this error. The original sentence mentioned the spectra that were physical, but then it was changed without changing the sense of the inequality. In the revised version, the text and the equation will be consistent.

L. 459-464: on the setting of "B_ph", I think in your study this setting is critical to compute the angular shape of the VSF and therefore the impact of the anisotropy of the remote sensing reflectance (e.g. BRDF).

We appreciate this comment. It is true that $B_{ph}$ is more important for BRDF applications than for others, i.e., algorithm development using reflectances at a fixed geometry. Initially, it might be thought as being critical, but in reality, it is not. The parameter that is more linked to the anisotropy is the backward lobe of $\beta / b_b$ (Sullivan and Twardowski 2009; Twardowski and Tonizzo 2018). In our synthetic dataset, the phase

functions are from the Fournier-Forand family. It can be shown that for all the Fournier-Forand family, $\beta / b_b$ is kept within a very narrow range of variability, which is also consistent with measured data (not shown). Therefore, given a reasonable $B_{ph}$ based on some other criteria, we can be confident that the anisotropy is well represented.

Nevertheless, we provided independent validation, or *closure*, for the $B_{ph}$ in our study using unique data by Whitmire et al. (2010), in Fig. 4. Such a validation has never been performed by any other paper presenting a synthetic dataset, and properly shows that our modelling of phytoplankton scattering is realistic, given all data available for check. Extended explanations will be given in the revised version.

Fig 5: x-label "C/N" very close to carbon to nitrate ratio….

We regret this potential confusion but since the variables are properly defined in the paper and there is no mention to either carbon or nitrogen, neither here nor in the development of other synthetic optical datasets, we keep such notation.

L. 704: "1300 angles", could you give the range (and increment) of the viewing zenith and azimuth angles

There must be a phrasing problem in the paragraph. The range and increment of the viewing and azimuth angles is indicated 700-704. We will try to make it clearer in the revised version.

L.748: letter "Psi" not defined

Indeed $\Psi$ was not defined before appearing. We will properly define it as the scattering angle the first time it is used.

L. 765: "… for a turbid water scenario", could you give the sediment (SPM) concentration

SPM is not a driving variable for our synthetic dataset. Instead, the chlorophyll concentration C and the non-algal particles concentration N are included. Here, $C = 16.9 \; mg \; m^{-3}$ and $N = 0.13 \; g \; m^{-3}$. We will replace the term "turbid" by "productive".

**References**

Berthon, J.-F., E. Shybanov, M. E. G. Lee, and G. Zibordi. 2007. Measurements and modeling of the volume scattering function in the coastal northern Adriatic Sea. Applied Optics **46:** 5189-5203.

Brando, V. E., and A. G. Dekker. 2003. Satellite hyperspectral remote sensing for estimating estuarine and coastal water quality. IEEE Transactions on Geoscience and Remote Sensing **41:** 1378-1387.

Brando, V. E., A. G. Dekker, Y. J. Park, and T. Schroeder. 2012. Adaptive semianalytical inversion of ocean color radiometry in optically complex waters. Applied Optics **51:** 2808-2833.

Bricaud, A., M. Babin, A. Morel, and H. Claustre. 1995. Variability in the chlorophyll-specific absorption coefficients of natural phytoplankton: Analysis and parameterization. J. Geophys. Res. **100:** 13321.

Casey, K. A. and others 2020. A global compilation of in situ aquatic high spectral resolution inherent and apparent optical property data for remote sensing applications. Earth Syst. Sci. Data **12:** 1123-1139.

Dierssen, H. M., R. A. Vandermeulen, B. B. Barnes, A. Castagna, E. Knaeps, and Q. Vanhellemont. 2022. QWIP: A Quantitative Metric for Quality Control of Aquatic Reflectance Spectral Shape Using the Apparent Visible Wavelength. Frontiers in Remote Sensing **3**.

IOCCG. 2006. Remote Sensing of Inherent Optical Properties: Fundamentals, Tests of Algorithms, and Applications, p. 1-122. *In* V. Stuart [ed.], Reports of the International Ocean-Colour Coordinating Group. International Ocean-Colour Coordinating Group, IOCCG.

Loisel, H., D. S. F. Jorge, R. A. Reynolds, and D. Stramski. 2023. A synthetic optical database generated by radiative transfer simulations in support of studies in ocean optics and optical remote sensing of the global ocean. Earth Syst. Sci. Data **15:** 3711-3731.

Mobley, C. D., L. K. Sundman, and E. Boss. 2002. Phase function effects on oceanic light fields. Applied Optics **41:** 1035.

Nechad, B. and others 2015. CoastColour Round Robin data sets: a database to evaluate the performance of algorithms for the retrieval of water quality parameters in coastal waters. Earth Syst. Sci. Data **7:** 319-348.

Park, Y.-J., and K. Ruddick. 2005. Model of remote-sensing reflectance including bidirectional effects for case 1 and case 2 waters. Applied Optics **44:** 1236-1249.

Pitarch, J., G. Volpe, S. Colella, R. Santoleri, and V. Brando. 2016. Absorption correction and phase function shape effects on the closure of apparent optical properties. Applied Optics **55:** 8618-8636.

Sullivan, J. M., and M. S. Twardowski. 2009. Angular shape of the oceanic particulate volume scattering function in the backward direction. Applied Optics **48:** 6811.

Tonizzo, A., M. Twardowski, S. McLean, K. Voss, M. Lewis, and C. Trees. 2017. Closure and uncertainty assessment for ocean color reflectance using measured volume scattering functions and reflective tube absorption coefficients with novel correction for scattering. Applied Optics **56:** 130-146.

Twardowski, M., and A. Tonizzo. 2018. Ocean Color Analytical Model Explicitly Dependent on the Volume Scattering Function. Applied Sciences **8:** 2684.

Valente, A. and others 2022. A compilation of global bio-optical in situ data for ocean colour satellite applications – version three. Earth Syst. Sci. Data **14:** 5737-5770.

Vandermeulen, R. A., A. Mannino, S. E. Craig, and P. J. Werdell. 2020. 150 shades of green: Using the full spectrum of remote sensing reflectance to elucidate color shifts in the ocean. Remote Sensing of Environment **247:** 111900.

Whitmire, A. L., W. S. Pegau, L. Karp-Boss, E. Boss, and T. J. Cowles. 2010. Spectral backscattering properties of marine phytoplankton cultures. Opt. Express **18:** 15073-15093.

Zibordi, G., and J. F. Berthon. 2024. Coastal Atmosphere & Sea Time Series (CoASTS) and Bio-Optical mapping of Marine optical Properties (BiOMaP): the CoASTS-BiOMaP dataset. Earth Syst. Sci. Data Discuss. **2024:** 1-33.

**A hyperspectral and multi-angular synthetic dataset for algorithm development in waters of varying trophic levels and optical complexity**

Jaime Pitarch, Vittorio Ernesto Brando

**Comments**

This paper presents a new synthetic data set linking apparent and inherent optical properties based on a very substantial set of radiative transfer simulations that are intended to provide comprehensive representation of optical water types found in nature. The purpose is to support ocean colour algorithm development and there is specific effort made to cover a wide range of sun sensor geometries, high spectral resolution and other important features. I am generally supportive of the effort and believe that the ambition of the work is significant. However, there are a couple of areas where I feel there are issues that might be either addressed or at least acknowledged before publication goes forward.

We thank the appreciation by the reviewer for the importance of this dataset and the amount of work that was put into it.

**Limitations of measured data sets:** One of the key themes of the paper is an ambition to better replicate the true range of variability found in nature. This is particularly emphasised with respect to oligotrophic waters which are reasonably claimed to be relatively under-sampled. In several sections, the authors point to existing field data sets and attempt to replicate all of the observed variability. Whilst this appears sensible on first inspection, I believe there is an underlying issue that needs to be considered. Essentially this boils down to the quality of field data. Any measurement is going to be subject to uncertainty and in many (most?) cases this uncertainty will become more significant as signal levels become smaller. This has been explored in some papers e.g. ref 1. Examples from the current manuscript that I think need to be considered include Figures 6 and 9 which both show apparently very strong variations in spectral slopes that just happen to coincide with signa levels dropping to very low levels.  Is this real variability or is it the result of poor quality fits caused by limited data quality when signals are very low?  Does it make sense to reproduce this level of variability in a synthetic data set if it is actually effectively noise and therefore potentially misleading? I think this at least needs to be considered.

This study relied on bio-optical data made publicly available in open access databases or single entries related to publications. Given the large amount of data that we gathered, we decided to apply stringent quality criteria. Specifically, for CDOM, it was required to follow an excellent fit to an exponential curve, thus ensuring that there were no issues with the spectral response or with the filtration. Most importantly, CDOM data had to be measured with Ultrapath devices, and the very long optical path should provide sufficient signal to noise ratio even in oligotrophic waters. Still, we acknowledge that there is an inevitable level of uncertainty related to in situ measurements.

There are also known issues with aspects of filter pad absorption measurements (Ref 2 pathlength amplification and baseline correction - the latter can also be an issue for CDOM absorption) that are not discussed but that could lead to significant discrepancies in observed data sets. These issues are effectively being baked into the training of this synthetic data set. The description of how these data were measured is lacking detail and I think there is scope to at least mention that there may be issues of this nature.

Processing differences among in situ datasets of filter pad data are not traced in the databases. That would require to go back to the original sources. It is, however, assumed that the practitioners, based on their experience, followed best practices. Indeed, most of these data come after the funding of projects by space agencies that involve related studies, and we believe that the groups that were involved were confident enough in the quality of the data before uploading. This said, the quality criteria to select valid filter pad

data was as stringent as for CDOM, so we are highly confident also here. But to keep the readers aware, we will include part of this comment in the manuscript.

**Oligotrophic under-sampling:** The authors make a significant play on extending coverage of oligotrophic waters that have been historically under-sampled. Whilst this is true, it remains the case that these waters have been sampled. I am concerned that Figure 9 appears to show at least a full order of magnitude of additional CDOM (ag440) range that has never been observed, even with Ultrapath CDOM sampling. I am perfectly happy to criticise measurement quality (see above) but I am a bit concerned about the justification for effectively inventing an additional decade of variability in this parameter? It is possible that community measurements have a lower limit that inhibits resolution of lower signals, but it is also potentially true that there is a background level of dissolved organic absorption that is a natural feature. I am not convinced that this aspect of the data set is as reliable as the paper currently suggests. Again, a more careful discussion of potential merit or otherwise would be advisable I think.

There is some misunderstanding here, which we aim at clarifying here and in the manuscript. $a_g(440)$ values ranging $2 \cdot 10^{-3} - 1 \cdot 10^{-2}$ were observed in South Pacific gyre during Biosope cruise as reported in Figure 15 of Bricaud et al. (2010), so we are not "effectively inventing an additional decade of variability in this parameter". Figure 9 in the manuscript is used for the construction of a remationship between parametrization of $S_g$ from $a_g(440)$, so we can predict the former from the latter. Only the CDOM data that passed the stringent quality control mentioned above are shown in the figure. It appears that those Biosope spectra were excluded for the parametrization of $S_g$: the text commenting Figure 9 will be updated to clarify the issue.

For the range of low CDOM values present in the synthetic dataset itself, we are in line with empirical data and another synthetic dataset (Loisel et al. 2023). To make it totally clear in the revised version, and clarify that the synthetic data set does not introduce an oligotrophic over-sampling, the lower-left panels in Figures 10 and 11 in the manuscript will be replotted showing $a_g(440)$ in the horizontal axes, and the related text will be updated accordingly to state that the synthetic dataset covers appropriately the $a_g(440)$ ranges of in situ and other synthetic datasets. The two panels are reported here as Figure R2.1. See how Loisel's dataset, the NOMAD dataset and our dataset have few points that down to about $a_g(440) \sim 3 \cdot 10^{-4} \ m^{-1}$.

[Figure]

Figure R2.1 Cross-relationship comparison for $a_g(440)$ and the $a_{ph}(440)/a_g(440)$ between the synthetic dataset and (left) various in situ datasets (right) other synthetic datasets.

**Parameterisation:** The paper takes considerable effort to describe and justify construction of the bio-optical model and other aspects which go into parameterising the Hydrolight runs. Inevitably there are decisions that need to be made and options discarded as a result. This is fine, but in several cases here various decisions are presented as inevitable when in fact alternative option could have been chosen. I would not ask for these decisions to be revered or for models to be reworked in addition - that would be unfair. However, I think it is possible for the authors to recognise that alternatives would be available and might also be legitimate options.

We will provide more context to justify the main decisions. Still, justifying all possible options would make the manuscript much lengthier than it already is. It is usually accepted as good practice to properly validate or justify the option that was taken.

For example, they have opted to use the a version of the Hydrolight input generation where they calculate backscattering from backscattering ratios applied to scattering coefficients rather than directly inputting backscattering SIOPs.

Yes and no. Phytoplankton backscattering is calculated from backscattering rations applied to scattering coefficients as the reviewer points out, but that has been the choice for all datasets (IOCCG 2006; Loisel et al. 2023; Nechad et al. 2015). Just for the principle of parsimony, we stuck to the usual procedure here. Still, we went one step beyond them because we were able to justify this parameterization based on the independent data in Figure 4 of the manuscript.

For non-algal properties, it was actually backscattering that was fixed, after knowing absorption (Figure 7 of the manuscript). We will revise the text in the related areas to see if we can provide some additional justification of the modelling choices.

I can point to a small number of papers where there have been efforts made to directly estimate thee parameters (refs 3 and 4) and which would have provided alternative options that could be considered. Again, I would like to emphasise that I am not looking for more work to be done here, just that there is a slightly less emphatic description of what is possible and available (or not), taking into account material that is not hard to find in the literature.

Also here, we will try to add some more context in the method description, without departing too much from the main topic, which is the dataset description.

**Validation:** The synthetic data set produces hyperspectral remote sensing reflectance spectra that may be of great value for algorithm development. However, it is unclear how representative the simulated spectra actually are? The discussion of the outputs very rapidly branches off into cluster analysis and consideration of geometric effects, but there is no real analysis of how representative the spectra are of natural distributions. I would like to see a comparison with existing measured data sets to get a sense of where there are overlaps and divergences that may or may not be of interest when considering value as an allegedly global data set. I would emphasise that I have no trouble with the quality of the simulated reflectance spectra per se – Hydrolight will produce essentially the right reflectance spectrum for whatever conditions you tell it to work with. However, the value of this synthetic data set is very much in its ability to cover the range of naturally occurring variation and I would like to see harder evidence that it does this e.g. for turbid coastal waters as well as more open coastal and oceanic conditions.

The reviewer has a point here. In the revised version, the reviewer will find strong evidence that the generated reflectances are in line with existing measured datasets in what regards absorption, chlorophyll concentration, total suspended matter concentration, plus an addition evaluation through a spectral quality

index. All in all, the validation exercises support our dataset as representative of a wide range of natural waters. We paste them here:

[Figure]

Figure R2.2 Upper plot: scatter plot between the apparent optical wavelength (Vandermeulen et al. 2020) and the NDI index: $NDI(492,665) = \frac{R_{rs}(665) - R_{rs}(492)}{R_{rs}(665) + R_{rs}(492)}$. Magenta lines: QWIP score (Dierssen et al. 2022) and error bars. Lower plot: histogram of the QWIP score, defined as the difference respect to the QWIP curve.

To generate Figure R2.2, we calculated the QWIP index by Dierssen et al. (2022) for our entire synthetic dataset. Such index aims at providing a quality estimate for a hyperspectral $R_{rs}$. QWIP was developed a large dataset of in situ $R_{rs}$, so this comparison is actually a comparison with real $R_{rs}$ data. In Dierssen et al. (2022), it is mentioned that values within the 0.2 margins have high similarity to real spectra measured in the field, which are all 5000 but 7 spectra. Still, these 7 spectra are close to the limit, and may simply contain some bio-optical characteristics, not present in the QWIP calibration dataset. This comparison, therefore, gives confidence in the quality of our dataset.

[Figure]

Figure R2.3 A scatter plot between the $R_{rs}$-generated $\chi$ index and the matched non-water absorption spectrum at 560 nm $a_{nw}(560)$. Black dots are from the synthetic dataset and coloured dots are from field data from various references (see text).

Figure R2.3 helps to assess the covariability of $R_{rs}$ and the absorption coefficient. A one-dimensional predictor $\chi$ is derived from an $R_{rs}$:

$$\chi = \log_{10}\left( \frac{R_{rs}(443) + R_{rs}(490)}{R_{rs}(560) + 5\frac{R_{rs}^2(665)}{R_{rs}(490)}} \right)$$

This $\chi$ index is matched to non-water absorption spectrum at 560 nm $a_{nw}(560)$. There are several open access, freely available in situ datasets that contain both measured variables matched together, such as Valente et al. (2022), Zibordi and Berthon (2024) and the Schaeffer, Mouw and Biosope datasets (Casey et al. 2020). Figure R2.3 clearly shows the excellent average overlap between our synthetic dataset and measured data. Different bio-optical characteristics produce slight deviations from the mean curve, indicating natural variability.

[Figure]

Figure R2.4 Chlorophyll concentration as a function of the maximum band ratio for OC4-type algorithms, for the synthetic dataset and for data in Valente et al. (2022) and Zibordi and Berthon (2024).

Figure R2.4 shows how a given chlorophyll concentration in the dataset relates to the generated $R_{rs}$ through an index that is used to estimate chlorophyll in the ocean:

$$MBR_{OC4} = \frac{\max\left[R_{rs}(443), R_{rs}(490), R_{rs}(510)\right]}{R_{rs}(560)}$$

From $R_{rs}$, we calculate the maximum band ratio $MBR_{OC4}$, an index known to be a good predictor for its good correlation to chlorophyll concentration (C) in oceanic waters, but also used for studying the consistency of a given dataset in all kinds of water (Nechad et al. 2015). Here, matched $MBR_{OC4}$ and chlorophyll concentration from two large in situ datasets are plotted (Valente et al. 2022; Zibordi and Berthon 2024), showing a good general overlap, though with some degree of differences among them, that are explainable due to different bio-optical characteristics of the seas sampled. Data from our dataset

generally agrees with the trend.

[Figure]

Figure R2.5 Total suspended matter concentration as a function of $R_{rs}(665)$, for the synthetic dataset and for data in Valente et al. (2022) and Zibordi and Berthon (2024).

The last comparison to real $R_{rs}$ data involves the relationship to the total suspended matter concentration (T), relevant for coastal and inland water, which usually show higher turbidities. Our dataset does not need T for its generation, but it can be estimated as T=N+0.07C, after Brando and Dekker (2003), where N is the concentration of non-algal particles. It is known that T covaries with $R_{rs}$ at long wavelengths, and 665 nm is commonly employed, due to the lesser disturbance by CDOM. Figure R2.5 shows that our dataset has a range of natural variability that includes that in in situ datasets (Valente et al. 2022; Zibordi and Berthon 2024), once more confirming the suitability of this new dataset for optical studies in all ranges of water.

The reader must note that for the new plots discussed above, the dot cloud amplitude in the in situ datasets is included in the synthetic dataset, meaning that the statistical treatment that was given to the inherent optical properties prior to radiative transfer simulations was such to ensure optical representativeness of many water types, as far as this plot is concerned.

**Final comment:** I have pointed to four references that are all from my own work. I am very uncomfortable doing this and I am NOT looking for these to be specifically referred to. They do, however, represent the basis for where my opinions have been shaped on these matters and where I believe we might have some philosophical differences that are not, however, insurmountable.

II would be more comfortable with a slightly less emphatic version of the paper that provides the reader with clear explanations of the decisions that were taken, but that notes that alternative options could have been taken in at least some cases.

The decisions that were taken were properly (although it may sound "emphatically") justified in the paper more than in any other similar paper before, to the limit that the preprint already has 59 pages. However,

without prejudice to our responses to previous comments of this review, we will try to provide some more background and guidance to the reader on the option choice.

I genuinely think the authors need to carefully consider the rationale for reproducing all of the observed variability, including measurement uncertainties, some of which are very significant indeed.

The rationale behind this study is that not accounting for the variability would lead to a dataset that would be too self-similar, as was the case for the Coastcolour dataset. The choice incorporating the spread in crossed relationships into the bio-optical modelling goes back to the IOCCG dataset, and we applied here the same principle, with the difference that now we have much more data and the ability to constrain some crossed relationships that could not be constrained before. Unfortunately, the discussion on uncertainty is very difficult, since IOP measurements never come with an uncertainty estimate. We acknowledge that a part of the data cloud spread is due to measurement and errors and mismatches, but the validation exercises show that the reflectance shows patterns and dispersion that agrees with measurements.

Ultimately I would be unlikely to use this synthetic data set as I would struggle to accept some of the decisions that have gone into producing it, but I can imagine it being welcomed by a significant part of the community more or less as is.

We regret this reluctance as the dataset is already being used to develop the bidirectional reflectance correction algorithm to be operationally implemented in EUMETSAT's processing chain of OLCI Level 2 data. The dataset itself is valuable for many other algorithm calibration or validation purposes, but also the research leading to its generation represents a relevant contribution.

As with all of these things, the expression caveat emptor pertains. I hope that these comments will help to encourage a slightly less emphatic description of the data set and encourage potential users to be mindful of where the limitations might still be found.

We appreciate that the reviewer took the time to help improve the manuscript. The reviewer will see enhanced validation and explanations in the revised version. *In dubio pro reo*.

**References**

McKee, D., R. Röttgers, G. Neukermans, V. Sanjuan Calzado, C. Trees, M. Ampolo-Rella, C. Neil and A. Cunningham Impact of measurement uncertainties on determination of chlorophyll-specific absorption coefficient for marine phytoplankton. J. Geophys. Res. Oceans: 119, 9013–9025, doi:10.1002/2014JC009909, 2014.

Lefering, I., R. Röttgers, R. Weeks, D. Connor, C. Utschig, K. Heymann, and D. McKee Improved determination of particulate absorption from combined filter pad and PSICAM measurements, Opt. Express 24, 24805-24823, 2016.

Bengil, F., D. McKee, S. T. Beşiktepe, V. S. Calzado, and C. Trees A bio-optical model for integration into ecosystem models for the Ligurian Sea, Prog. Oceanography 149, 1-15, 2016.

Lo Prejato M., McKee D. (2023) Optical Constituent Concentrations and Uncertainties Obtained for Case 1 and 2 Waters From a Spectral Deconvolution Model Applied to In Situ IOPs and Radiometry. Earth and Space Science, 10 (12), art. no. e2022EA002815. DOI: 10.1029/2022EA002815

**References**

Brando, V. E., and A. G. Dekker. 2003. Satellite hyperspectral remote sensing for estimating estuarine and coastal water quality. IEEE Transactions on Geoscience and Remote Sensing **41:** 1378-1387.
Bricaud, A., M. Babin, H. Claustre, J. Ras, and F. Tièche. 2010. Light absorption properties and absorption budget of Southeast Pacific waters. Journal of Geophysical Research: Oceans **115**.

Casey, K. A. and others 2020. A global compilation of in situ aquatic high spectral resolution inherent and apparent optical property data for remote sensing applications. Earth Syst. Sci. Data **12:** 1123-1139.

Dierssen, H. M., R. A. Vandermeulen, B. B. Barnes, A. Castagna, E. Knaeps, and Q. Vanhellemont. 2022. QWIP: A Quantitative Metric for Quality Control of Aquatic Reflectance Spectral Shape Using the Apparent Visible Wavelength. Frontiers in Remote Sensing **3**.

IOCCG. 2006. Remote Sensing of Inherent Optical Properties: Fundamentals, Tests of Algorithms, and Applications, p. 1-122. *In* V. Stuart [ed.], Reports of the International Ocean-Colour Coordinating Group. International Ocean-Colour Coordinating Group, IOCCG.

Loisel, H., D. S. F. Jorge, R. A. Reynolds, and D. Stramski. 2023. A synthetic optical database generated by radiative transfer simulations in support of studies in ocean optics and optical remote sensing of the global ocean. Earth Syst. Sci. Data **15:** 3711-3731.

Nechad, B. and others 2015. CoastColour Round Robin data sets: a database to evaluate the performance of algorithms for the retrieval of water quality parameters in coastal waters. Earth Syst. Sci. Data **7:** 319-348.

Valente, A. and others 2022. A compilation of global bio-optical in situ data for ocean colour satellite applications – version three. Earth Syst. Sci. Data **14:** 5737-5770.

Vandermeulen, R. A., A. Mannino, S. E. Craig, and P. J. Werdell. 2020. 150 shades of green: Using the full spectrum of remote sensing reflectance to elucidate color shifts in the ocean. Remote Sensing of Environment **247:** 111900.

Zibordi, G., and J. F. Berthon. 2024. Coastal Atmosphere & Sea Time Series (CoASTS) and Bio-Optical mapping of Marine optical Properties (BiOMaP): the CoASTS-BiOMaP dataset. Earth Syst. Sci. Data Discuss. **2024:** 1-33.